# Structured Representation Learning with Locally Linear Embeddings and Adaptive Feature Fusion

**Somjit Nath**  *somjit.nath@mail.mcgill.ca*
*Department of Electrical and Computer Engineering*
*McGill University*
*Mila – Quebec AI Institute*

**Jackson J Cone**  *jackson.cone@ucalgary.ca*
*Department of Psychology*
*Department of Biomedical Engineering*
*University of Calgary*
*Canada Research Chair in Computational Behavioural Neuroscience (Tier II)*

**Derek Nowrouzezahrai**  *derek@cim.mcgill.ca*
*Department of Electrical and Computer Engineering*
*McGill University*
*Mila – Quebec AI Institute*
*Canada CIFAR AI Chair*

**Samira Ebrahimi Kahou**  *samira.ebrahimikahou@ucalgary.ca*
*Department of Electrical and Software Engineering*
*Schulich School of Engineering*
*University of Calgary*
*Mila – Quebec AI Institute*
*Canada CIFAR AI Chair*

**Reviewed on OpenReview:** *https://openreview.net/forum?id=p7p3iuah0G*

## Abstract

Neuroscientific research has revealed that the brain encodes complex behaviors by leveraging structured, low-dimensional manifolds and dynamically fusing multiple sources of information through adaptive gating mechanisms. Inspired by these principles, we propose a novel reinforcement learning (RL) framework that encourages the disentanglement of dynamics-specific and reward-specific features, drawing direct parallels to how neural circuits separate and integrate information for efficient decision-making. Our approach leverages locally linear embeddings (LLEs) to capture the intrinsic, locally linear structure inherent in many environments—mirroring the local smoothness observed in neural population activity—while concurrently deriving reward-specific features through the standard RL objective. An attention mechanism, analogous to cortical gating, adaptively fuses these complementary representations on a per-state basis. Experimental results on benchmark tasks demonstrate that our method, grounded in neuroscientific principles, improves learning efficiency and overall performance compared to conventional RL approaches, highlighting the benefits of explicitly modeling local state structures and adaptive feature selection as observed in biological systems.

## 1 Introduction

Reinforcement learning (RL) has achieved remarkable success in domains such as games (Mnih et al., 2015; Silver et al., 2017) and robotic control (Yuan et al., 2025; Tang et al., 2024; Sun et al., 2021). Despite

these achievements, one of the longstanding challenges in RL is the development of state representations that not only support reward maximization but also faithfully capture the underlying dynamics of the environment. Many traditional methods optimize a single objective that integrates both aspects, often resulting in representations that are tuned predominantly to immediate reward signals and may neglect the intrinsic structure that governs state transitions(Mnih et al., 2015; Schulman et al., 2017; Haarnoja et al., 2018).

A promising alternative is to augment the RL framework with unsupervised representation learning techniques (Jaderberg et al., 2016). In simulated robotic domains, the primary focus of our experiments, the state transitions are governed by underlying physical principles, meaning that movement between states exhibits *predictable, and well-defined local* relationships. Locally linear embedding (LLE) (Roweis & Saul, 2000) is a well-established technique for preserving local neighborhood structures and has found widespread use in dimensionality reduction and manifold learning (Tenenbaum et al., 2000; Belkin & Niyogi, 2003). However, its potential for decomposing the state representation into dynamics-specific features has not yet been fully explored within the RL context.

In this paper, we propose a dual-representation framework for RL that utilizes LLE to explicitly extract dynamics-specific features while reward-specific features are learned concurrently through conventional RL objectives. By encouraging the disentanglement of these two aspects, our method enables the agent to form a more comprehensive internal model of the environment, a capability that is particularly beneficial in the simulated robotic domains we study. To effectively combine these complementary representations, we use a self-attention mechanism (Vaswani et al., 2017). This mechanism dynamically evaluates the relative importance of the dynamics-specific features extracted via LLE and the reward-specific features learned from the RL objective on a per-state basis. Furthermore, the attention weights can be visualized as attention maps, which serve as a diagnostic tool by revealing at which states the agent relies more heavily on one set of features over the other. These visuals can be invaluable for understanding the decision-making process of RL agents in complex, simulated environments. From a neuroscientific standpoint, this dual-representation design is not arbitrary. The dynamics-specific branch, which captures locally linear structure via LLE, plays the role of a dedicated dynamics manifold analogous to motor and association cortices that encode smooth, low-dimensional trajectories of neural activity. In parallel, the reward-specific branch mirrors reinforcement-sensitive circuits (e.g., dopaminergic pathways and basal ganglia) that track task outcomes and value. The attention-based fusion module then acts as a cortical gating mechanism that dynamically combines these streams, closely reflecting how the brain selectively routes and integrates information across specialized subsystems during decision-making.

Thus, our architectural choices are not only motivated by engineering convenience, but are explicitly aligned with how biological systems organize and combine dynamics-specific and reward-specific information. Neuroscience studies have shown that the brain mediates complex functions like movement, navigation, decision-making, and cognition within low-dimensional manifolds of population activity, where local relationships between neural states are preserved and adapted over time (Sabatini & Kaufman, 2024; Gallego et al., 2018; Langdon et al., 2023). Furthermore, the brain employs context-sensitive gating and attention mechanisms—mediated by the frontal cortex to selectively amplify relevant features and flexibly integrate multiple information streams (Martinez-Trujillo, 2022; Bichot et al., 2015). These insights motivate the development of RL agents that not only maximize reward but also explicitly capture the underlying dynamics and structure of their environment, much like their biological counterparts.

Traditional RL methods often conflate reward-driven and dynamics-specific information, resulting in representations that may neglect the intrinsic structure governing state transitions (Mnih et al., 2015; Schulman et al., 2017; Haarnoja et al., 2018). In contrast, the brain maintains specialized subsystems for representing environmental dynamics, reinforcement, and cognition, allowing for more flexible and robust learning. Inspired by this, we propose a dual-representation framework that tries to disentangle these two aspects, leveraging LLE; a technique that preserves local geometric relationships, akin to the local smoothness observed in neural population activity and fuses them with reward-specific features via an attention mechanism reminiscent of cortical gating. By grounding our approach in neuroscientific principles, we aim to endow RL agents with the ability to form structured and adaptable internal models. Our experiments in simulated robotic

domains, where physical laws enforce predictable state transitions, demonstrate that this biologically inspired separation and adaptive fusion of features leads to improved learning efficiency and overall performance.

Our experimental evaluation across a range of simulated robotic tasks shows that our framework not only improves learning efficiency but also enhances overall performance compared to conventional RL methods. The insight gained from the attention maps further suggests that the proposed method provides a useful balance between dynamics and reward representations, adapting flexibly to the requirements of different phases of the task.

The remainder of this paper is organized as follows. Section 2 reviews related work on state representation learning in RL and discusses unsupervised methods such as LLE. Section 3 details our proposed framework and the associated training procedures. In Section 4, we present our experimental results, and Section 5 concludes the paper with a discussion of our findings and directions for future research.

## 2 Related Works

**Manifold Learning.**  Manifold learning techniques—including locally linear embedding (Roweis & Saul, 2000) and its variants (Ghojogh et al., 2020)—have been widely used for dimensionality reduction and visualization in static datasets. These methods excel at preserving local neighborhood structures under the assumption that high-dimensional data lie on a low-dimensional manifold. However, applying such techniques directly in an online RL setting is challenging due to dynamic state distributions and non-stationary objectives. Our work extends the idea of local linearity by incorporating adaptive update mechanisms (with dynamic loss thresholds and window size considerations) to extract robust features from RL trajectories. We integrate the LLE features into the policy learning pipeline and fuse them with reward-driven features via self-attention, enabling a more flexible and interpretable representation.

Relatedly, several approaches explicitly model environment structure to support generalization and decision making. For instance, rSLDS (Glaser et al., 2020) models switching linear dynamical regimes with latent discrete states, DeepSynth (Hasanbeig et al., 2021) encodes structure in a discrete automaton that guides RL, and SWMPO (Cano et al., 2025) builds a neurosymbolic world model that explicitly represents dynamics structure. In contrast, our method does not infer discrete regimes, synthesize automata, or learn an explicit world model; instead, it provides a lightweight *representation-level inductive bias* that preserves local neighborhood geometry in the embedding (via LLE) and combines it with reward-driven features through adaptive fusion. This perspective is complementary to explicit structure-learning methods and can potentially be combined with them.

**Attention Mechanisms.**  Attention has emerged as a powerful tool in deep learning, with applications ranging from natural language processing to computer vision. In RL, early attempts such as the work by Manchin et al. demonstrated that self-supervised attention can improve task performance by highlighting task-relevant features. More recent approaches, for example, the self-reinforcement attention mechanism for tabular learning (Amekoe et al., 2023), have sought to incorporate attention to facilitate interpretability in structured data. Our method leverages self-attention not merely to reweight features but to perform a learned fusion between two distinct representations: the unsupervised embedding capturing local geometry and the reward-specific features. This explicit separation and subsequent adaptive fusion distinguishes our approach from methods that encode attention within a monolithic network architecture and offers improved performance.

**Auxiliary Losses and Decoupled Representation Learning.**  The efficacy of auxiliary losses in RL has been demonstrated in recent works which automatically search for or hand-design additional objectives to improve representation quality (Lange et al., 2024).

Self supervised learning (SSL) and contrastive methods have significantly advanced representation learning, particularly in RL tasks that rely on high-dimensional sensory inputs, such as images. Techniques like Contrastive Unsupervised Representations for Reinforcement Learning (CURL) (Srinivas et al., 2020), Momentum Contrast for Unsupervised Visual Representation Learning (MoCo) (He et al., 2020) and Simple Framework for Contrastive Learning of Visual Representations (SimCLR) (Chen et al., 2020) leverage

contrastive learning to enhance feature discrimination by ensuring representations of similar states are pulled together while representations of different states are pushed apart. Similarly, data augmentation methods such as Data-regularized Q-learning (DrQ) (Kostrikov et al., 2021; Yarats et al., 2021) and Reinforcement Learning with Augmented Data (RAD) (Laskin et al., 2020) apply transformations to raw visual inputs, improving sample efficiency and robustness. Contrastive dynamics-focused methods such as Deep Bisimulation for Control (DBC) (Zhang et al., 2021) attempt to enforce invariances across transitions, but these methods remain agnostic to the precise local geometry of the state space.

However, these approaches are primarily designed for image-based RL, where raw pixels must be processed into meaningful state representations. In environments where state representations already exhibit intrinsic structure, such as physics-based robotic tasks, representation learning need not infer structure from scratch but can instead exploit inherent local relationships present in the state space. Our method capitalizes on this by leveraging LLE a technique designed to preserve local geometric consistency. Rather than relying on augmentation or contrastive objectives, our approach grounds representation learning in the underlying system dynamics, ensuring that learned embeddings remain faithful to the physical principles governing state transitions.

Furthermore, while contrastive-based SSL methods require extensive negative sampling or data augmentation to form meaningful latent spaces, our LLE approach builds representations directly from local neighborhood information in the state space. This eliminates the need for explicit augmentation and instead allows representations to organically reflect the underlying transition dynamics. Additionally, by integrating LLE with reward-driven features through an attention mechanism, our model dynamically prioritizes the most relevant aspects of the state at different stages of training. This adaptive representation fusion results in improved sample efficiency, particularly in structured state spaces where local smoothness is a beneficial inductive bias. Moreover, while these approaches often involve joint optimization of the RL objective and auxiliary tasks, they risk interfering with the learning dynamics if the auxiliary signal does not align well with the main task. Instead, our method decouples the learning of local structure from reward-based learning; the LLE module is updated according to its own convergence criteria, and its output is fused with the primary features via a self-attention layer. This decoupling mitigates adverse interactions between auxiliary tasks and the policy, and in our experiments, leads to better sample efficiency compared to standard auxiliary approaches.

## 3 Methodology

Our methodological design directly operationalizes the neuroscientific principles outlined in the Introduction 1. Concretely, we instantiate a dynamics-specific pathway that captures locally linear structure (reflecting low-dimensional neural manifolds), a reward-specific pathway that optimizes behavior (reflecting reinforcement-sensitive circuits), and an adaptive fusion mechanism that gates information between them (reflecting cortical attention and gating). The following subsections detail how each of these components is implemented.

### 3.1 Our Framework

Our dual-representation framework as illustrated in Figure 1 divides the state representation into two complementary components: **dynamics-specific features** and **reward-specific features**. The dynamics-specific branch uses LLE to explicitly capture the locally linear structure in state transitions, while the reward-specific branch learns features from the standard RL objective. In our architecture, the two feature sets are first concatenated and then processed via a self-attention layer to yield a refined representation that guides action selection.

### 3.1.1 Dynamics-specific Representation via Locally Linear Embedding

We leverage LLE (Roweis & Saul, 2000) to capture the intrinsic, locally linear structure of the state transitions. Given an input state $x \in \mathbb{R}^D$, for each state $x_i$ we first identify its $K$ nearest neighbors $\{x_{i_1}, x_{i_2}, \ldots, x_{i_K}\}$ selected from a small temporal window around it, e.g., $x_{t-K/2}, \ldots, x_{t-1}, x_{t+1}, \ldots, x_{t+K/2}$.

Figure 1: Overview of the dual-representation framework. The input state is processed in parallel through the dynamics-specific branch using LLE to extract local structure and the reward-specific branch via conventional RL learning. After concatenation, a self-attention layer adaptively fuses the features, producing a refined representation for action selection.

**Step 1: Computing neighbor reconstruction weights** $w_{ij}$**.** The reconstruction weights $\{w_{ij}\}$ are found by minimizing:

$$L_W = \sum_i \left\| x_i - \sum_{j \in \mathcal{N}(i)} w_{ij}\, x_j \right\|^2, \quad \text{s.t.} \sum_{j \in \mathcal{N}(i)} w_{ij} = 1. \tag{1}$$

**Step 2: Updating the Low-Dimensional Embeddings** $z$**.** Once the weights $w_{ij}$ are computed, the low-dimensional embeddings $z_i^{\text{LLE}} \in \mathbb{R}^d$ (with $d \ll D$) are obtained by minimizing:

$$L_z = \sum_i \left\| z_i^{\text{LLE}} - \sum_{j \in \mathcal{N}(i)} w_{ij} z_j^{\text{LLE}} \right\|^2. \tag{2}$$

Following standard LLE, we compute neighbor reconstruction weights $w_{ij}$ in observation space via Eq. (1), and then enforce neighborhood preservation in the learned embedding by reusing the same weights to reconstruct $z_i^{\text{LLE}}$ from its neighbors, i.e., Eq. (2). In our framework, $z^{\text{LLE}}(x)$ is produced by a parameterized encoder $e_\theta(x)$, and $\theta$ is updated to minimize $L_z$, rather than computing a closed-form LLE embedding via eigendecomposition. In our approach, both $L_W$ and $L_z$ are minimized using a similar iterative procedure detailed in the next subsection.

From a neuroscientific perspective, this procedure can be viewed as learning a stable latent manifold on which neighboring states share similar local linear reconstructions. This mirrors empirical findings that neural population activity evolves along smooth trajectories within a shared, low-dimensional manifold, where nearby neural states encode closely related behaviors or contexts.

**Note:** The dynamics-specific representation is state-to-embedding: it maps a single state/observation $x_t$ to $z_t^{\text{LLE}}$, and we do not learn an explicit transition model $p(x_{t+1} \mid x_t, a_t)$. Here, "dynamics-specific" refers to the fact that the embedding is shaped by local structure induced by the environment's evolution: the LLE constraint is applied to states sampled from policy rollouts and uses trajectory-local (temporal) neighbors, encouraging embeddings that are locally consistent along feasible transitions in the replay buffer, rather than being shaped directly by reward gradients.

Moreover, the LLE neighbor weights $w_{ij}$ are not stored globally as persistent parameters. Instead, they are computed within minibatches sampled from the replay buffer (or periodically), resulting in $O(BK)$ storage per LLE update, where $B$ is the minibatch size and $K$ is the number of neighbors. Neighbors are selected directly in observation space using trajectory-local temporal neighboring states, which avoids an expensive global kNN search over the full replay buffer and substantially reduces neighbor-search overhead.

### 3.1.2 Reward-specific Representation via RL

In parallel, the reward-specific branch learns features $z^{\text{Rew}}(s)$ from the input state $s$ using a conventional deep RL network (e.g., actor-critic or Q-network). This branch is trained using the standard RL loss functions that drive cumulative reward maximization.

### 3.1.3 Attention-Based Fusion via Self-Attention

To seamlessly integrate the two feature types, we first concatenate the dynamics-specific features $z^{\mathrm{LLE}}(s)$ and the reward-specific features. Let the dynamics-specific (LLE) encoder produce $z^{\mathrm{LLE}}(s) \in \mathbb{R}^d$ and the reward-specific encoder produce $z^{\mathrm{Rew}}(s) \in \mathbb{R}^d$. We form the combined representation by *feature-based concatenation* (concatenation along the feature/channel dimension):

$$f_{\mathrm{cat}}(s) = \mathrm{concat}\left(z^{\mathrm{LLE}}(s),\, z^{\mathrm{Rew}}(s)\right) \in \mathbb{R}^{2d}.$$

We use the convention $z^{\mathrm{LLE}}, z^{\mathrm{Rew}} \in \mathbb{R}^d$ and $f_{\mathrm{cat}} \in \mathbb{R}^{2d}$, so $f_{\mathrm{cat}}$ is a vector-valued feature representation.

This concatenated vector is then processed by a self-attention layer (Vaswani et al., 2017), which computes refined representations by learning interactions between the different feature components:

$$f(s) = \mathrm{SelfAttn}(f_{\mathrm{cat}}(s)).$$

The self-attention mechanism allows the agent to assess, on a per-state basis, which aspects of the dynamics-specific and reward-specific features are most critical for decision-making. Moreover, the computed attention weights can be visualized as attention maps, offering insights into the usefulness of the learned features. In biological terms, this fusion layer acts like a cortical gating mechanism that flexibly prioritizes either dynamics-related or reward-related features depending on the current context. Just as frontal cortical circuits gate which inputs are amplified or suppressed during complex behavior, our self-attention module adaptively routes information between the LLE manifold and reward-specific representations to construct a task-relevant state embedding.

### 3.2 Interleaved Training Procedure and Detailed Algorithm

Training alternates between reward-driven SAC updates and structure-driven LLE updates. The overall loop is: (i) collect experience into a replay buffer, (ii) update policy/critic using standard SAC losses, and (iii) periodically update the dynamics-specific encoder using the LLE objectives. This interleaving avoids optimizing the LLE objective to convergence at every iteration, thus reducing compute.

**Step 1: Data collection.** At each environment step, we sample an action using the current policy, execute it in the environment, and store the transition $(x_t, a_t, r_t, x_{t+1})$ in the replay buffer, where $x_t$ denotes the input state vector.

**Step 2: SAC update (reward-specific branch).** We sample a minibatch of transitions from the replay buffer and perform standard SAC updates. This updates the policy and critic parameters (and the reward-specific encoder / fusion parameters via backprop through the policy/critic computation graph). During this step, the dynamics-specific encoder is not optimized by the SAC objective.

**Step 3: Periodic LLE update (dynamics-specific branch).** Every *lle_update_interval* or $B$ steps, we perform an LLE update using a minibatch of $B$ states $\{x_i\}_{i=1}^B$ sampled from the replay buffer. For each $x_i$, we construct a neighbor set $\mathcal{N}(i)$ using trajectory-local temporal neighbors in observation space. We then compute scalar neighbor reconstruction weights $w_{ij}$ by minimizing Eq. 1, and update the dynamics-specific encoder parameters by minimizing Eq. 2. During this LLE step, only the dynamics-specific encoder parameters are updated by the LLE losses.

**Early stopping and minibatch optimization.** Each LLE update operates on a minibatch of size *lle_batch_size* sampled from the replay buffer and uses early stopping, either when the loss decreases by less than a threshold $\epsilon$ consecutively 5 times or when a maximum number of optimization steps is reached.

**Reconstruction regularizer (Eq. 3).** To prevent representation collapse and ensure that $z^{\mathrm{LLE}}$ remains information-preserving, we regularize the dynamics-specific embedding with a reconstruction objective:

$$L_{\mathrm{rec}} = \sum_{i=1}^B \left\| x_i - A z_i^{\mathrm{LLE}} \right\|_2^2. \tag{3}$$

**Shapes:** $x_i \in \mathbb{R}^D$, $z_i^{\text{LLE}} \in \mathbb{R}^d$, and $A \in \mathbb{R}^{D \times d}$; hence $A z_i^{\text{LLE}} \in \mathbb{R}^D$ and the squared error is scalar. The neighbor coefficients $w_{ij}$ in Eqs. 1–2 are distinct from the decoder parameters $A$ in Eq. 3: $w_{ij}$ are per-sample scalar weights used to preserve local neighborhood structure, while $A$ parameterizes an independent reconstruction regularizer.

**Reference pseudocode.** Algorithm 1 provides pseudocode for the full interleaved loop and Algorithm 2 details the LLE update.

### 3.3 Neuroscientific Foundations of Locally Linear Embeddings and Adaptive Feature Fusion

**Locally Linear Embeddings** Recent research has shown that the motor cortex exhibits smoothly evolving neural population activity, where states at nearby time points remain highly correlated, while activity diverges over longer time scales (Sabatini & Kaufman, 2024). This finding suggests that neural computations occur within a structured, low-dimensional manifold that preserves local relationships between neural states. LLEs operate under a similar premise: they unfold high-dimensional data while maintaining local neighborhood consistency, ensuring that neighboring states in the original space remain close in the learned representation.

Furthermore, neural studies indicate that a common latent manifold supports diverse motor behaviors, with slight adjustments to the underlying dynamics yielding different movements (Gallego et al., 2018). This property aligns with the functionality of LLE, where representations are learned in a way that preserves local geometry while allowing flexible modifications for different reinforcement learning tasks. In our approach, LLE enables structured feature extraction by capturing intrinsic local smoothness, much like how the brain encodes adaptable motor dynamics.

Another key parallel lies in reward-driven modulation of motor control. Dopaminergic systems selectively reinforce neural pathways associated with positive outcomes, effectively tagging regions of the motor manifold that contribute to successful behavior (Gadagkar et al., 2016; Bromberg-Martin et al., 2010). Our reinforcement learning framework mirrors this process by learning which areas of the LLE representation are more predictive of reward, reinforcing their importance in policy optimization. As a result, our model dynamically adjusts feature prioritization, akin to biological motor learning mechanisms.

In summary, both biological neural systems and LLE-based representations leverage locally constrained dynamics to generate structured behaviors. The ability to maintain local relationships while permitting adaptive modifications enables efficient learning, whether in cortical circuits or RL architectures. Taken together, these observations support our decision to implement a dedicated dynamics pathway based on LLE. By enforcing locally linear reconstruction within a low-dimensional embedding, our model instantiates a computational analogue of the neural manifolds observed in motor and association cortices, while the reinforcement learning branch selectively assigns credit to regions of this manifold that are behaviorally advantageous.

**Adaptive Feature Fusion** Neuroscientific research has established that the brain selectively amplifies relevant features for learning through gating mechanisms mediated by the frontal cortex. This gating plays a crucial role in distinguishing important environmental signals from distracting or interfering information, allowing for efficient adaptation to complex behaviors (Anton-Erxleben et al., 2009; Egner & Hirsch, 2005). In our framework, self-attention mechanisms serve a similar function by dynamically reweighting features based on their relevance to decision-making.

Furthermore, the ability to learn complex, potentially interfering representations in neural systems relies on context-specific gating within the frontal cortex (Barbosa et al., 2023; Johnston et al., 2007; Shallice et al., 2008). In our framework, self-attention provides an analogous gating mechanism, ensuring that critical aspects of the learned representation are retained while filtering out less relevant components. This adaptive feature selection prevents overfitting to noisy patterns and allows for efficient task-specific learning.

Within our architecture, this perspective is realized by the self-attention module that consumes both LLE and reward-specific features. Rather than treating attention as an engineering trick, we interpret it as a computational model of frontal cortical gating: a mechanism that learns, from experience, when to emphasize stable dynamics-driven structure and when to rely more heavily on reward-driven signals. This

tight correspondence between biological gating and our adaptive fusion mechanism is central to the improved robustness and interpretability we observe in our experiments. By dynamically adjusting representation focus in response to reward signals, self-attention functions as a computational parallel to frontal cortical gating in the brain.

## 4 Experimental Results

We evaluate our dual-representation approach within the Soft Actor–Critic (SAC) framework (Haarnoja et al., 2018), which is well regarded for its sample efficiency and stability in robotic applications. We employ widely used robot simulation environments because their realistic physical dynamics render the locally linear assumption underlying our LLE module especially, valid physical laws, enforce predictable state transitions that our method can exploit. We ran all experiments with 10 random seeds and report the mean and standard error. The code for reproducing the results is open-sourced [1]. Additionally, additional results, hyperparameters used for all the environments, and a compute comparison between the baselines are in Appendix Section B and Tables 1 & 2 respectively.

### 4.1 Baselines

All experiments were conducted under the SAC framework (Haarnoja et al., 2018). While our LLE-based representation learning framework is compatible with any RL algorithm, we selected Soft Actor-Critic (SAC) because it is widely regarded as the state-of-the-art off-policy algorithm for continuous control tasks, including those in Robosuite, which served as a benchmark for our experiments. In addition to vanilla SAC, we compare our proposed SAC-LLE method against several auxiliary-loss baselines (Lange et al., 2024):

- **SAC-Recon**: SAC with an auxiliary state reconstruction loss.
- **SAC-Next**: SAC with a next-state prediction loss conditioned on the current action.
- **SAC-Reward**: SAC augmented with an immediate reward prediction loss.
- **SAC-SPR**: SAC with self predictive representations (Schwarzer et al., 2020) (implemented without data augmentation given our low-dimensional state).
- **SAC-DBC**: A dynamics-based contrastive baseline following (Zhang et al., 2021) that learns representations by enforcing consistency across dynamics-preserving state transitions.
- **SAC-LLE (Joint)**: A variant that jointly learns LLE-based features with reward-specific features without an explicit fusion layer.

**Note:** We found that "SAC" implementations and training schedules differ substantially across codebases, which can materially affect absolute returns in Robosuite. For Robosuite, we additionally report results using the SAC implementation and update schedule aligned with the Robosuite reference repository to enable a more direct comparison with reported curves in the Robosuite paper. For Dexterous Gym, this Robosuite SAC variant did not yield learning progress without substantial retuning (near-zero returns across tasks), so we report DexGym results using the stable DexGym SAC configuration throughout. In all cases, comparisons between our method and baselines are performed under the same protocol within a domain. The exact hyperparameters are in Table 1 in Appendix.

### 4.2 Environments

**Robosuite Experiments** Robosuite (Zhu et al., 2025) is a modular simulation framework built on top of the MuJoCo physics engine, designed for benchmarking robotic manipulation tasks. It provides a diverse set of environments featuring various robot arms (e.g., Panda, Sawyer) and objects, enabling the study of complex manipulation skills under realistic physical constraints. Each environment specifies the robot model, object properties, and task-specific goals, such as lifting, stacking, or assembling objects. The state space typically includes joint positions, velocities, end-effector poses, and object states, while the action space consists of continuous control commands for the robot actuators. Robosuite tasks used in our experiments include:

---

[1] https://github.com/Somjit77/lle-rl

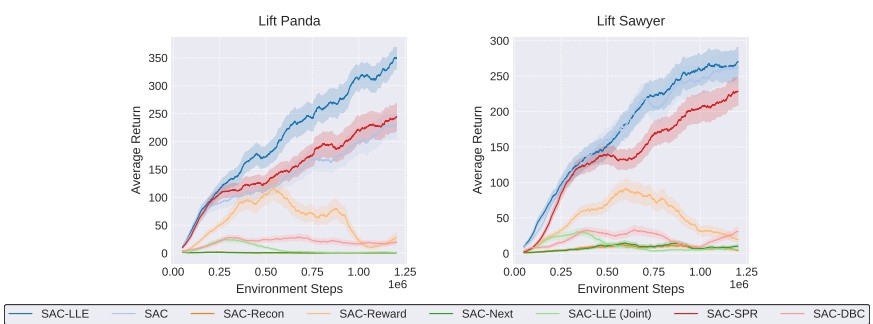

Figure 2: Learning curves on RoboSuite Lift. Curves show mean training return across 10 seeds as a function of environment steps; shaded regions indicate standard error across seeds.

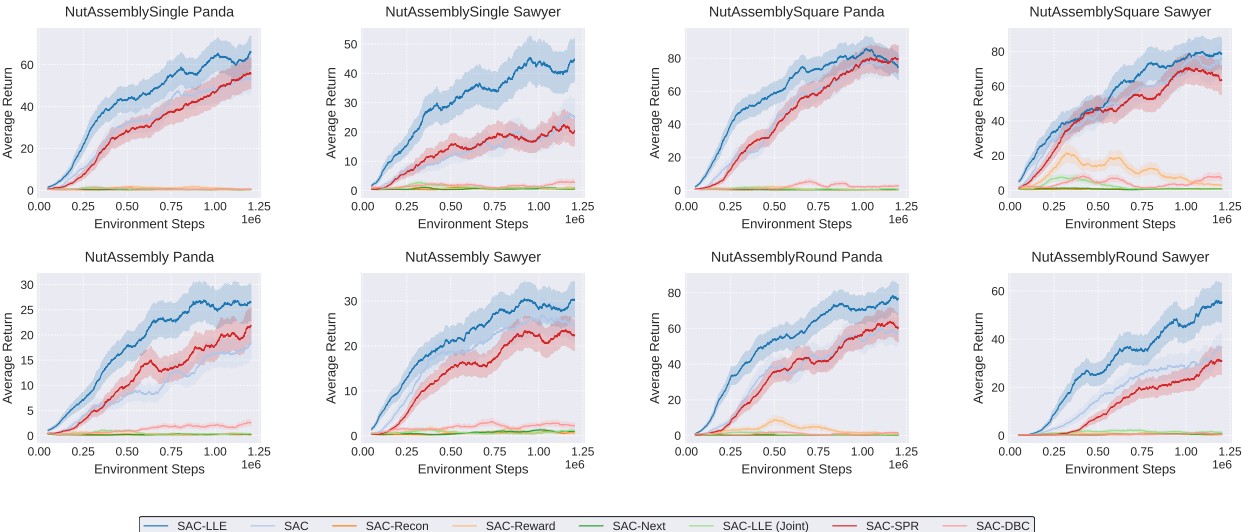

Figure 3: Learning curves on NutAssembly Lift. Curves show mean training return across N seeds as a function of environment steps; shaded regions indicate standard error across seeds.

- **Lift**: The robot must grasp and lift an object from a table.
- **NutAssembly**: The robot assembles nuts onto pegs, with variants requiring different shapes or assembly strategies.
- **PickPlace**: The robot picks up objects and places them at target locations, with multiple object types (e.g., Can, Bread, Milk, Cereal).
- **Stack**: The robot stacks objects in a specified order.

Observations are provided as low-dimensional state vectors, and rewards are shaped to encourage task completion and efficient manipulation.

**Dexterous Gym Experiments**   Dexterous Gym (Charlesworth & Montana, 2021) is a suite of challenging dexterous manipulation environments based on the Shadow Hand robot. It focuses on tasks that require fine motor skills, such as grasping, rotating, and throwing objects. The environments simulate a 24-DoF anthropomorphic hand interacting with various objects (Pen, Block, Egg), each with unique physical properties and manipulation challenges. Tasks include:

- **CatchOverarm**: The hand must catch an object thrown overhand.
- **CatchUnderarm**: The hand must catch an object thrown underhand.

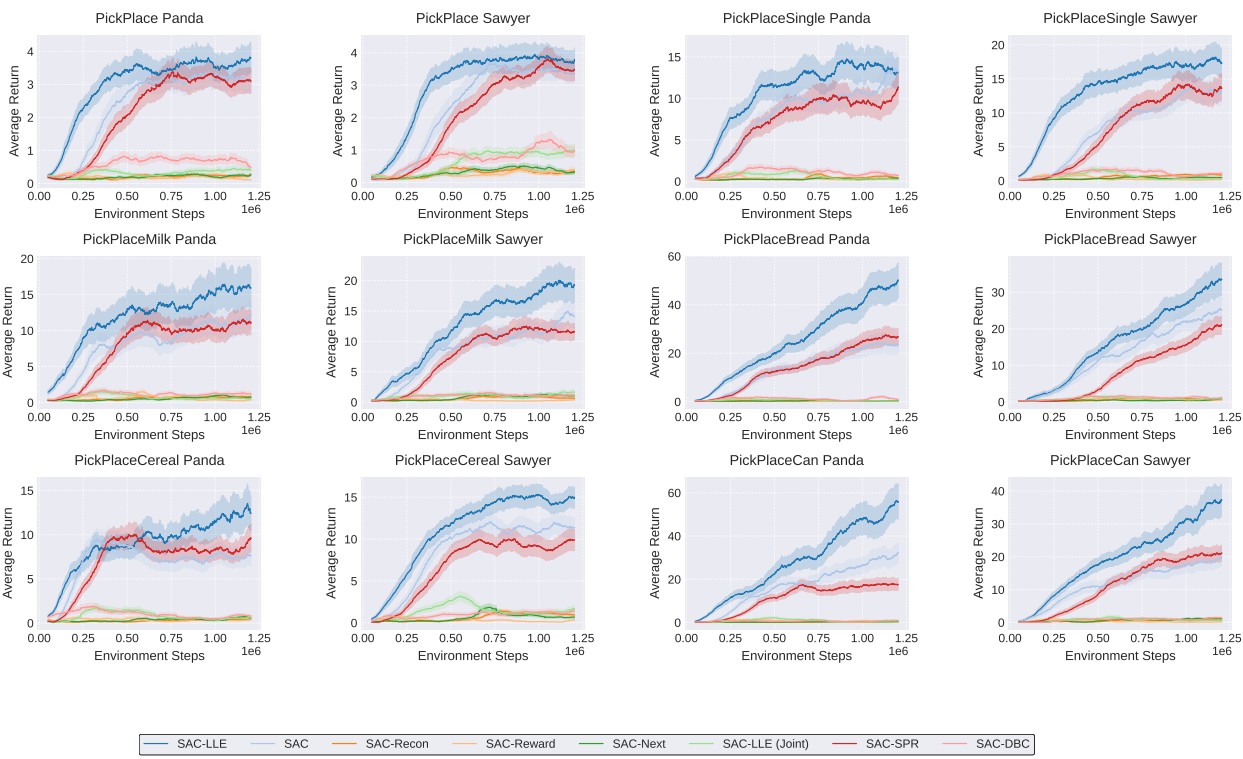

Figure 4: Learning curves on RoboSuite PickPlace. Curves show mean training return across N seeds as a function of environment steps; shaded regions indicate standard error across seeds.

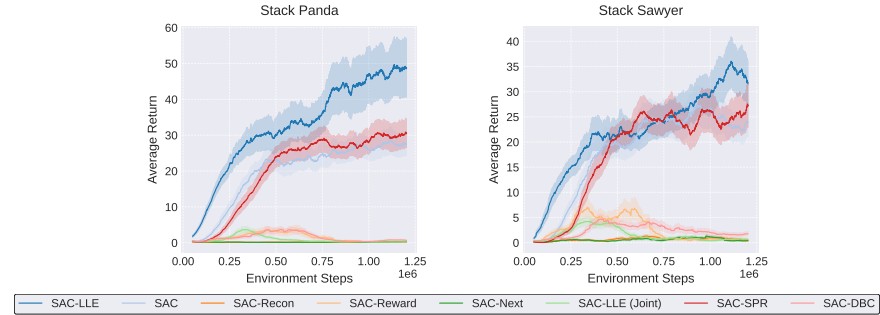

Figure 5: Learning curves on RoboSuite Stack. Curves show mean training return across N seeds as a function of environment steps; shaded regions indicate standard error across seeds.

The observation space includes joint angles, velocities, fingertip positions, and object states, while the action space consists of continuous joint torques. Rewards are typically sparse, given for successful task completion, making these environments particularly demanding for reinforcement learning algorithms.

## 4.3 Performance Gains

Our results indicate that SAC-LLE consistently outperforms these baselines, including the dynamics-based contrastive approach (SAC-DBC), which still falls short of our method, thus highlighting the benefit of explicitly modeling local geometry rather than relying solely on contrastive invariances. Notably, the relatively poor performance of SAC-LLE (Joint) suggests that forced joint learning may over-constrain the state space, inhibiting effective reward feature extraction. In contrast, learning LLE features separately and fusing them adaptively (SAC-LLE) produces significant performance improvements. This approach is more *structured*

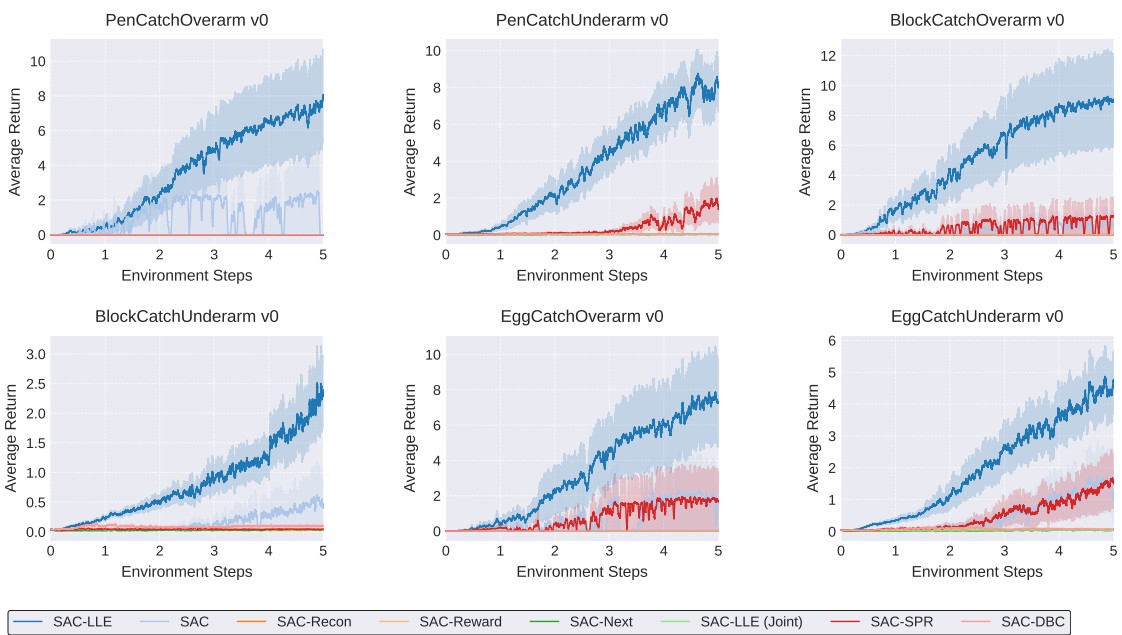

Figure 6: Learning curves on Dexterous Gym. Curves show mean training return across 10 seeds as a function of environment steps; shaded regions indicate standard error across seeds.

and *biologically inspired* compared to less targeted auxiliary losses. It avoids the pitfalls of purely data-driven representation learning by leveraging known geometric properties inherent to the environment, which guides the feature learning into a more meaningful space.

We observed that auxiliary losses designed for high-dimensional inputs such as images often contribute little improvement when applied to already rich, vectorized robotic states. Many standard auxiliary tasks, including next state prediction or reward prediction, are geared towards learning representations from unstructured inputs. By contrast, our approach constrains the learned feature space using locally linear embeddings (LLE) and encourages separation between dynamics-specific from reward-specific features, introducing a biologically motivated inductive bias that significantly enhances learning performance. This explains the comparatively poor performance of other baselines in our evaluated domains and underscores the unique contribution of our method.

## 4.4 Visualization of the Attention Maps

To gain insight into how our dual-representation model adapts its internal focus over the course of an episode, we analyze the attention maps produced by the self-attention layer for the Lift task on the Panda robot. We capture the model's attention at three representative stages of a rollout episode.

At the beginning of an episode (Figure 7a), the attention map shows a pronounced focus on the reward features (indices 0–128) with very little activation in the LLE feature range (indices 128–256). This indicates that the agent initially relies on explicit reward cues to guide its early decision-making. As we progress to the middle (Figure 7b) and the late stages (Figure 7c) of the episode, a notable shift occurs: distinct activation patterns emerge predominantly on the left-hand side of the map. These leftward patterns suggest that the initially separate reward and LLE features are becoming integrated. In other words, while the reward features continue to drive the agent's focus on high-value cues, they increasingly incorporate the subtle contextual and geometric relationships captured by the LLE features. This integration likely aids in refining the agent's perception by ensuring that the decision-making process is not solely driven by immediate rewards but is also informed by a robust internal representation of the scene's local structure.

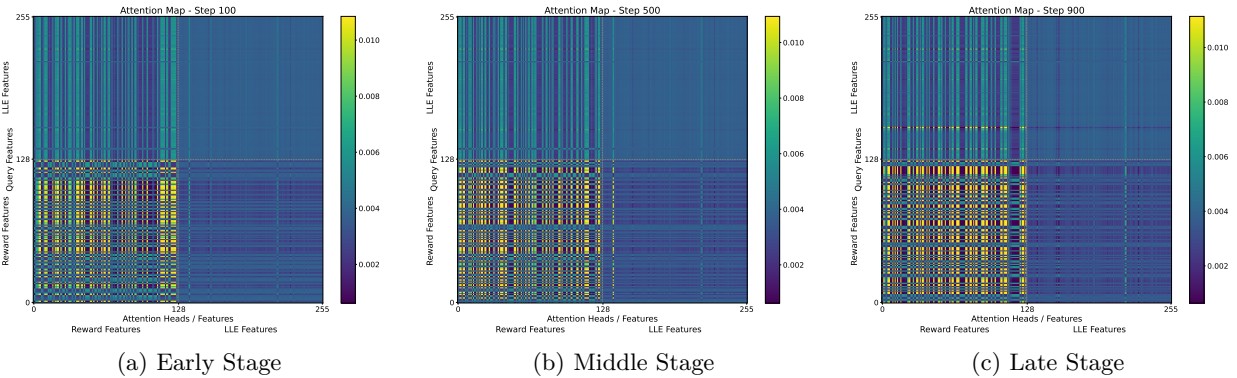

(a) Early Stage          (b) Middle Stage          (c) Late Stage

Figure 7: Evolution of the attention maps for the Lift task on the Panda robot. In the early stage (left), the attention is broadly distributed; in the middle (center), it begins to focus on specific regions; and by the late stage (right), the attention becomes highly concentrated on key features.

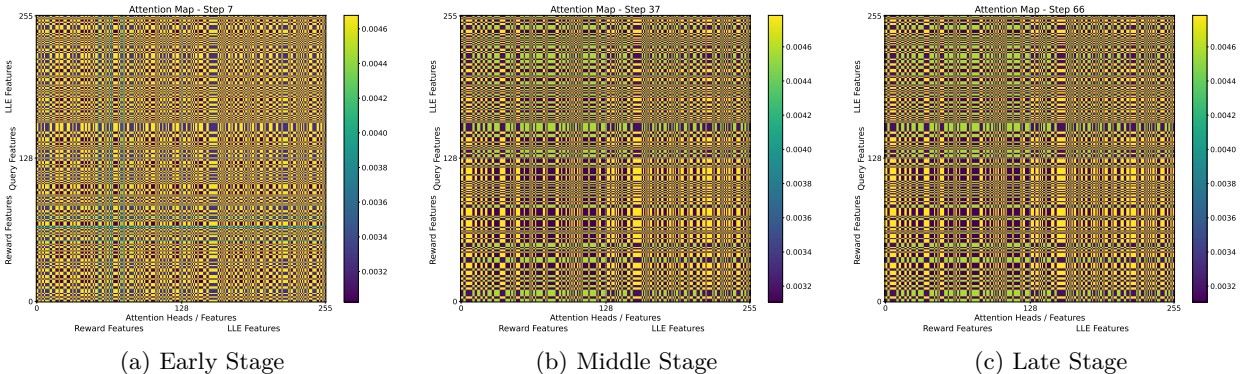

(a) Early Stage          (b) Middle Stage          (c) Late Stage

Figure 8: Evolution of the attention maps for the EggCatchUnderarm task on Dexterous Gym Suite. For Dexterous Gym environments, the LLE features contribute equally almost everywhere, further highlighting their importance.

This evolution in attention distribution suggests that as the episode unfolds, the network gradually shifts its focus from a wide array of low-level cues to a more refined and targeted feature set that is most informative for action selection.

In contrast, for the Dexterous Gym environments, the attention behavior is markedly different. As shown in Figure 8, the LLE feature range exhibits consistently strong activation across all stages of the episode—early, middle, and late. Rather than gradually integrating LLE information, the model appears to depend on these features from the very beginning. This persistent utilization indicates that, in dexterous manipulation tasks with rich contact dynamics and high-dimensional observations, the locally linear structure encoded by the LLE representation provides essential cues throughout the entire rollout. The network thus leverages LLE features not as supplementary context but as a primary, continuously accessed component of its internal representation.

This contrast highlights an important property of our dual-representation model: while attention over reward and LLE features evolves over time in simpler manipulation tasks, the more complex Dexterous Gym settings require and consistently draw upon the geometric and contextual information embedded in the LLE representation from start to finish.

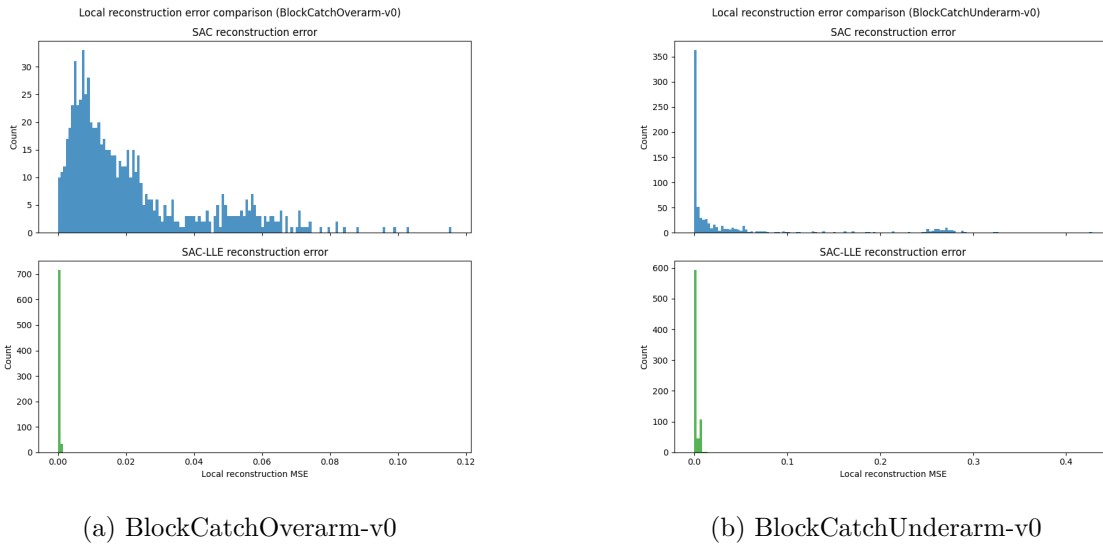

(a) BlockCatchOverarm-v0          (b) BlockCatchUnderarm-v0

Figure 9: Local reconstruction error comparison on BlockCatch tasks. We plot the distribution of per-state local reconstruction MSE (Eq. 3) for SAC and SAC-LLE over evaluation rollouts. In both environments, SAC-LLE concentrates near zero error, indicating stronger local structure preservation in the learned embedding.

## 4.5 Direct evidence of neighborhood preservation.

Beyond attention-map interpretability, we provide a direct quantitative measure of local structure preservation using the reconstruction objective. Concretely, for each point we find its $K$ nearest neighbors in the original observation space, solve a least-squares reconstruction of that point's embedding from the neighbors' embeddings, and report the reconstruction MSE in embedding space on 10 held-out evaluation rollouts across 10 seeds and compare SAC vs. SAC-LLE.

Figure 9 shows that SAC-LLE yields reconstruction errors concentrated near zero, whereas SAC exhibits a substantially heavier tail with larger errors. This indicates that the LLE-regularized representation preserves locally consistent structure more reliably in practice, providing direct support for our claim that the LLE constraint encourages neighborhood-preserving embeddings.

## 4.6 Ablation: Adaptive Feature Fusion vs. Static Concatenation

In our architecture, self-attention acts as an explicit, state-conditional fusion operator between the LLE features and the reward features: the two $d$-dimensional features are concatenated and then adaptively reweighted to produce a fused representation. While an actor/critic network operating on a static concatenation could in principle learn an implicit gating function, this pushes the burden of feature weighting into the downstream MLP and conflates fusion with policy capacity. By performing fusion explicitly, self-attention provides a stronger inductive bias for state-dependent feature selection.

Beyond performance, attention also provides interpretability: the resulting attention weights can be inspected to quantify which branch/features are emphasized in different regions of the state space (as visualized in the attention maps in Section 4.6). A pure concatenation does not naturally expose an equivalent per-state mixing mechanism. To isolate the contribution of the fusion mechanism, four variants are compared while keeping the underlying two-stream representation fixed: (i) *SAC-LLE (Self-Attention)*, the proposed model; (ii) *Concat-only*, which feeds $f_{\text{cat}}$ directly to the policy/critic; (iii) *MLP Fusion*, which replaces attention with a lightweight MLP-based fusion; and (iv) *Scalar Gate*, which uses a single learned gate to blend the two streams.

Figure 10 (a) shows that self-attention fusion achieves the strongest learning performance on Lift (Panda), while static concatenation underperforms, indicating that a learnable fusion operator is beneficial beyond

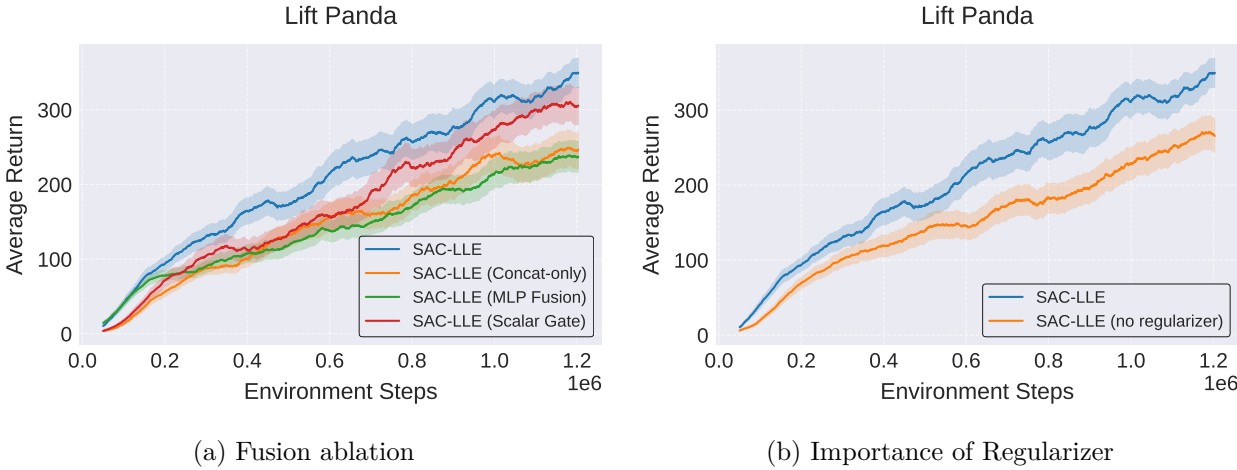

(a) Fusion ablation            (b) Importance of Regularizer

Figure 10: Ablations on Lift (Panda). (a) Fusion mechanism: SAC-LLE with self-attention fusion vs fusion alternatives that preserve the same two-stream representation (Concat-only, MLP Fusion, Scalar Gate). (b) Reconstruction regularizer: SAC-LLE with vs. without the Eq. (3) reconstruction term. Shaded regions indicate standard error across 10 seeds; returns shown are training returns.

simply increasing representational capacity via concatenation. The Scalar-gate variant partially recover performance, suggesting that adaptive fusion is important, with self-attention providing the most effective state-dependent gating in this setting.

### 4.7 Ablation: Effect of the Reconstruction Regularizer

Equation (3) introduces a reconstruction regularizer on the LLE branch. While the neighborhood preservation objective (Eqs. 1–2) enforces that embeddings respect locally linear neighborhood relationships, the reconstruction term encourages the learned embedding to remain information-preserving and helps avoid degenerate solutions (e.g., overly lossy or collapsed embeddings). This section isolates the contribution of the reconstruction regularizer.

Two variants are compared while keeping all other components fixed (SAC training protocol, two-stream architecture, fusion mechanism, neighbor construction, and update schedule): (i) *SAC-LLE*, which uses the LLE neighborhood objective together with the reconstruction regularizer, and (ii) *SAC-LLE (no regularizer)*, which removes the regularizer while retaining the neighborhood preservation objective.

Figure 10 (b) shows that removing the reconstruction regularizer degrades learning performance on Lift (Panda): *SAC-LLE* achieves higher average return than *SAC-LLE (no regularizer)* over the training horizon, indicating that the reconstruction regularization contributes materially to the performance gains by stabilizing and regularizing the learned representation.

## 5 Conclusion

In this work, we introduced a dual-representation framework for reinforcement learning that is explicitly motivated by neuroscientific principles. By leveraging locally linear embedding, which mirrors the brain's use of low-dimensional, locally linear manifolds, and integrating these with reward-specific features through a self-attention mechanism analogous to cortical gating, our approach enables adaptive feature fusion on a per-state basis. This biologically inspired design results in improved learning efficiency, enhanced overall performance, and increased understanding of useful features in simulated robotic domains. Experimental results demonstrate that modeling local state structure—particularly in environments governed by physical laws—provides tangible benefits over conventional RL approaches that conflate dynamics and reward information. Visualization of attention maps further offers valuable insights into the agent's decision-making process, revealing how

the importance of different feature types shifts throughout the episode, much like the dynamic adaptation observed in neural circuits. Overall, our approach highlights the value of decoupling and adaptively fusing distinct aspects of state representation, paving the way for more robust RL agents.

**Limitations and Future Research**   While our proposed dual-representation framework demonstrates promising results in structuring RL representations, we acknowledge several important limitations that should be considered. Our approach relies on the key assumption that environment dynamics exhibit locally linear structures in the state space. This assumption is well-founded in many robotic and physics-based domains where transitions between states follow smooth, physically consistent trajectories. However, in highly complex, chaotic, or visually-rich environments (e.g., raw pixel inputs), this local linearity may not hold, thus limiting the applicability of our method. Extending the framework to model non-linear or high-dimensional visual dynamics remains an important direction for future research.

While our current study evaluates the proposed LLE-based auxiliary loss exclusively on simulated environments, we chose these domains because their realistic physics and dynamical properties serve as close approximations to real robotic control tasks. These simulators inherently capture the local linearity assumptions underpinning our method, providing a rigorous and controlled testbed. Future research may extend this framework to broader classes of environments and explore integration of LLE and adaptive feature fusion to higher-dimensional state spaces like images, further bridging the gap between artificial and biological intelligence.

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

# A  Algorithm Description

---

**Algorithm 1** Overall Training Procedure for Dual-Representation RL Agent

---

1: **Input:** Replay buffer, initial parameters, *lle_batch_size*, *lle_update_interval*
2: **for** each training iteration **do**
3:     Sample a mini-batch of transitions $(s, a, r, s')$ from the replay buffer.
4:     Compute reward-specific features $z_{\text{RL}}(s)$ and calculate the RL loss $L_{\text{RL}}$.
5:     Compute initial dynamics-specific features $z_{\text{LLE}}(s)$ using the current LLE module.
6:     Update $z_{\text{LLE}}(s)$ using Equation 3 to ensure feature fidelity.
7:     Concatenate features and apply self-attention to obtain the fused representation:

$$f(s) = \text{SelfAttn}\left([z_{\text{LLE}}(s); z_{\text{RL}}(s)]\right)$$

8:     Update network parameters by performing a gradient descent step on $L_{\text{RL}}$.
9:     **if** training iteration   mod  *lle_update_interval* $= 0$ **then**
10:         Invoke LLE_UPDATE procedure.

---

**Algorithm 2** LLE_UPDATE Procedure

---

1: **Input:** Replay buffer, *lle_batch_size*; maximum steps $N_{\max}^W$ for $W$; maximum steps $N_{\max}^z$ for $z$; tolerances $\epsilon_W$ , $\epsilon_z$
2: Sample a mini-batch $B$ of size *lle_batch_size* from the replay buffer.
3: **Update $W$ (Equation 1)**
4: Compute initial loss $L_W^{\text{prev}} = L_W(B)$.
5: Set $\text{count}_W \leftarrow 0$.
6: **for** $k = 1$ **to** $N_{\max}^W$ **do**
7:     Compute current loss $L_W^{\text{curr}} = L_W(B)$.
8:     Update $W$ using a gradient descent step on $L_W$.
9:     **if** $L_W^{\text{prev}} - L_W^{\text{curr}} < \epsilon_W$ **then**
10:         $\text{count}_W \leftarrow \text{count}_W + 1$.
11:     **else**
12:         $\text{count}_W \leftarrow 0$.
13:     Set $L_W^{\text{prev}} \leftarrow L_W^{\text{curr}}$.
14:     **if** $\text{count}_W \geq 5$ **then**
15:         **break**.
16: **Update $z$ (Equation 2)**
17: Compute initial loss $L_z^{\text{prev}} = L_z(B)$.
18: Set $\text{count}_z \leftarrow 0$.
19: **for** $k = 1$ **to** $N_{\max}^z$ **do**
20:     Compute current loss $L_z^{\text{curr}} = L_z(B)$.
21:     Update $z$ using a gradient descent step on $L_z$.
22:     **if** $L_z^{\text{prev}} - L_z^{\text{curr}} < \epsilon_z$ **then**
23:         $\text{count}_z \leftarrow \text{count}_z + 1$.
24:     **else**
25:         $\text{count}_z \leftarrow 0$.
26:     Set $L_z^{\text{prev}} \leftarrow L_z^{\text{curr}}$.
27:     **if** $\text{count}_z \geq 5$ **then**
28:         **break**.

---

# B    Additional Results

## B.1    Robosuite Protocol Sensitivity

In the main text, we report Robosuite results using an SAC implementation and training protocol aligned with the Robosuite reference codebase to enable a more direct comparison with previously reported learning curves. During early experimentation (and in an earlier draft), we also evaluated our method and baselines using our default SAC training pipeline (the same we used for DexGym, with default horizon of 1000 as set by the latest version of Robosuite). We observed that absolute returns on Robosuite can vary substantially across SAC implementations and update schedules and horizon length.

For transparency, we include these Robosuite learning curves obtained under the default SAC protocol in this appendix. These results are provided to illustrate protocol sensitivity and to document the standard experimental setting. Importantly, within each protocol setting, all methods (baselines and SAC-LLE) are trained and evaluated consistently.

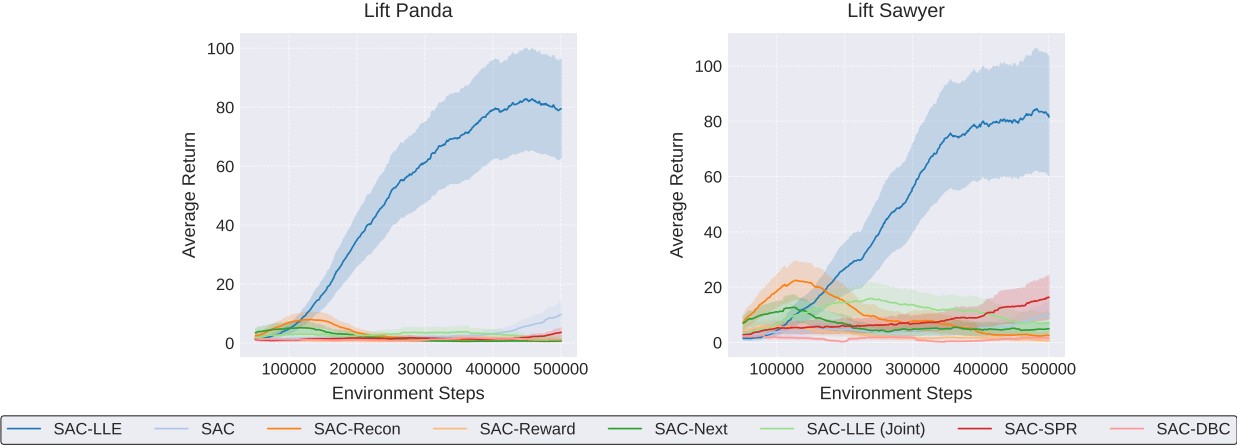

Figure 11: Learning curves for Lift tasks in Robosuite under the default SAC protocol. Shaded regions indicate standard error across 10 seeds; returns shown are training returns.

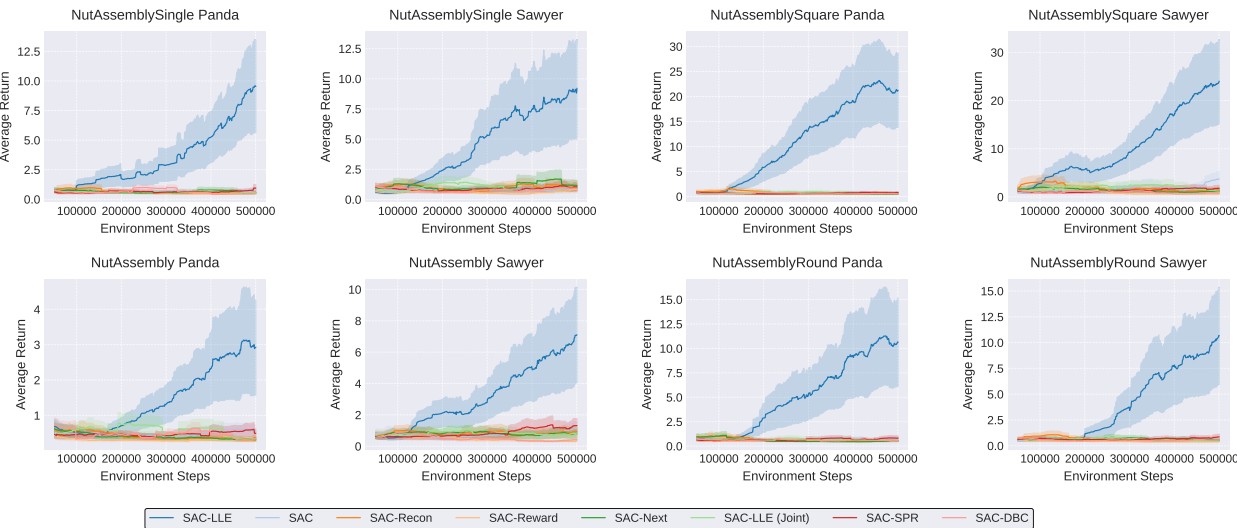

Figure 12: Learning curves for NutAssembly tasks in Robosuite under the default SAC protocol. Shaded regions indicate standard error across 10 seeds; returns shown are training returns.

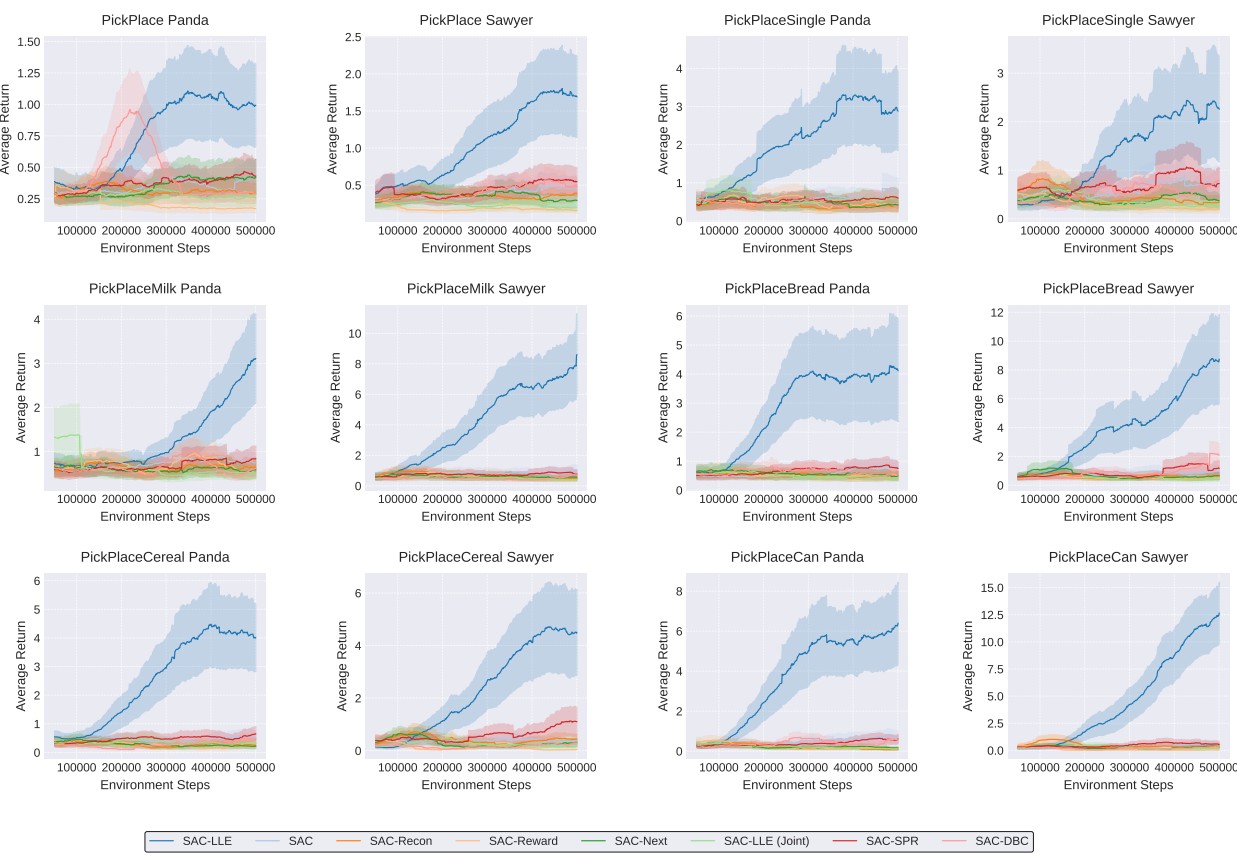

Figure 13: Learning curves for PickPlace tasks in Robosuite under the default SAC protocol. Shaded regions indicate standard error across 10 seeds; returns shown are training returns.

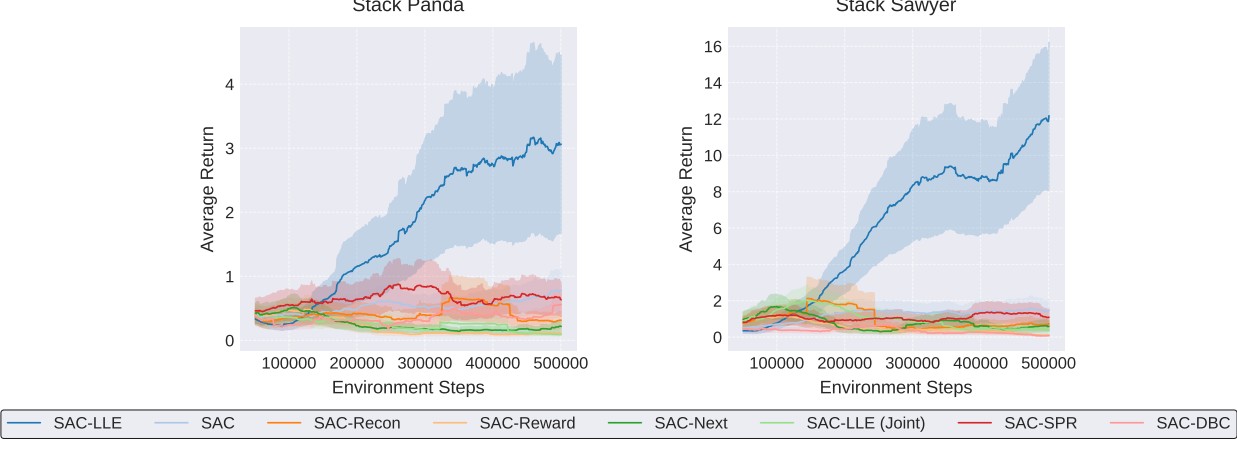

Figure 14: Learning curves for Stack tasks in PickPlace under the default SAC protocol. Shaded regions indicate standard error across 10 seeds; returns shown are training returns.

## B.2 Hyperparameter Sensitivity

In order to understand how various hyperparameters affect our dual-representation approach, we performed a series of ablation studies on the Lift task using the Panda robot. In our dynamic update procedure for the Locally Linear Embeddings (LLE) module, we have several critical hyperparameters:

1. **Loss Reduction Threshold for the Low-Dimensional Representation:** This threshold determines when to stop updating the low-dimensional embedding (i.e., $z$) based on the absolute loss reduction between successive gradient steps.

2. **Loss Reduction Threshold for the Weight Matrix:** Likewise, this parameter sets the stopping criterion for updating the reconstruction weight matrix (i.e., $W$).

3. **Local Window Size:** The number of nearest neighbors used to compute the local linear reconstruction weights. Adjusting this value affects how much local context is incorporated.

4. **Learning Rate for Updating $z$:** The step size for updating the low-dimensional representation. This needs to be carefully tuned to ensure stable yet expressive representation learning.

5. **Learning Rate for Updating $W$:** The step size for updating the reconstruction weight matrix, which must be set appropriately so that the local structure is captured effectively.

Figure 15 presents the sensitivity analysis for each of these hyperparameters. Each subfigure shows the average policy return as a function of the respective parameter value.

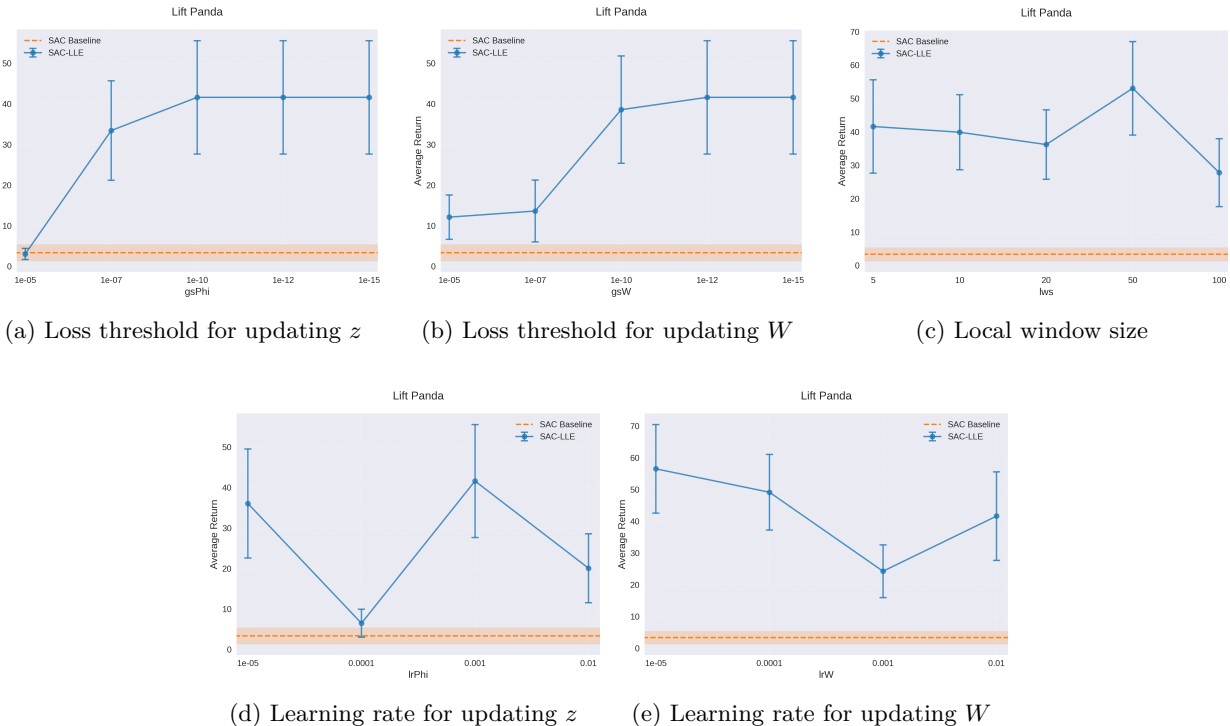

(a) Loss threshold for updating $z$    (b) Loss threshold for updating $W$    (c) Local window size

(d) Learning rate for updating $z$    (e) Learning rate for updating $W$

Figure 15: Ablation studies on key hyperparameters for the Lift task on the Panda robot. The plots depict how changing each parameter influences the average policy return.

Our experimental results indicate that there is an optimal range for each key hyperparameter. In particular, if the loss reduction thresholds for either $z$ or $W$ are set too high, the update process may stop prematurely, leading to under-trained representations. Conversely, setting these thresholds too low results in unnecessary computations with diminishing returns. Similarly, the choice of the local window size and the learning rates for updating $z$ and $W$ plays a critical role in balancing stability and expressiveness in our representation learning.

These findings emphasize the importance of careful hyperparameter tuning in our dual-representation framework and provide valuable insights for future exploration, including potential adaptive update strategies.

## C Hyperparameters & Compute Comparison

| Hyperparameter | DexGym | Robosuite |
|---|---|---|
| Local window size | 5 | 5 |
| LLE batch size | 2048 | 1024 |
| LLE learning rate ($W$) | 1e-2 | 1e-2 |
| LLE learning rate ($z_{\mathrm{LLE}}$) | 1e-3 | 1e-3 |
| Loss Reduction threshold ($W$) | 1e-15 | 1e-15 |
| Reduction threshold ($z_{\mathrm{LLE}}$) | 1e-10 | 1e-10 |
| Buffer size | 1e6 | 1e6 |
| Discount factor ($\gamma$) | 0.99 | 0.99 |
| Target smoothing coefficient ($\tau$) | 0.005 | 0.005 |
| Batch size | 256 | 128 |
| Learning starts | 5,000 steps | 3,300 steps |
| Train frequency (env steps) | 1 (every step) | 2,500 |
| Gradient steps per train loop | 1 | 1,000 |
| Policy learning rate | 3e-4 | 1e-3 |
| Q-network learning rate | 1e-3 | 5e-4 |
| Policy frequency | 2 | 1 |
| Target network frequency | 1 | 5 |
| Entropy coefficient ($\alpha$) | 0.2 (autotune enabled) | 0.2 (autotune enabled) |
| Hardware | A100 GPUs | H100 GPUs |

Table 1: Key Hyperparameters used for all the environments

| Algorithm | Training Steps per Second |
|---|---|
| SAC | 34 |
| SAC (with reconstruction) | 32 |
| SAC (SPR) | 27 |
| SAC (LLE) | 25 |

Table 2: Training steps per second for SAC variations on A100 GPU

