# OpenReview forum: "Structured Representation Learning with Locally Linear Embeddings and Adaptive Feature Fusion"
_TMLR — Accepted by TMLR_

### Review · Reviewer_rvtF · 2025-12-31

**Summary Of Contributions:**

The contributions are as follows:

- A RL framework that encourages the disentanglement of dynamics-specific and reward-specific features
- Evaluation of the RL framework on simulated dynamical systems (Robosuite and Dexterous Gym)

The authors claim that the framework is grounded in neuroscientific principles and "improves learning efficiency and overall performance compared to conventional RL approaches, highlighting the benefits
of explicitly modeling local state structures and adaptive feature selection as observed in biological systems."

**Audience:**

Yes

**Audience Explanation:**

Yes. Learning and leveraging structure within the RL process is an important, well-established problem and active area of research.

**Broader Impact Concerns:**

There are no ethical implications of this work that would require adding a Broader Impact Statement.

**Claims And Evidence:**

No

**Claims Explanation:**

Important aspects of the manuscript require clarification before it can be said the claims are supported by evidence:

1. Is it the case that $z_i^{\text{LLE}} = \sum_{j \in \mathcal{N}(i)} w_i z_j$? If so, perhaps this could be explicitly stated, as the computation of $z_i^\text{LLE}$ is core to the framework and is also not unambiguously described in the appendix. Otherwise, what is the relationship of Equation 2 and $z_i^{\text{LLE}}$?
2. You claim the proposed framework learns "dynamics-specific features". However, the embeddings seem to only involve state-wise information, and not information about dynamics, because the algorithm does not seem to leverage temporal relationships. Is this the case? If so, would you please clarify what you mean by "dynamics-specific features"?
3. You write, "realistic physical dynamics render the locally linear assumption underlying our LLE module especially, valid physical laws, enforce predictable state transitions that our method can exploit." Please clarify what you mean and explain why that is the case, particularly given Question 2 above.
4. (In equation 1) What is the domain of the indexed variable $x_i$?
5. (In 3.1.1) Similarly, where are the k-nearest neighbors chosen from?
6. It is quite surprising that all the baselines perform so poorly. For example, if you consult Figure 4 of the robosuite paper, the SAC-based Panda systems generally achieve more than 200 reward after 500 epochs, whereas your baselines reach less than 20 and your proposed system reaches 100. Would you clarify the details of your experiments and how the results are to be interpreted, particularly in relationship with the setup in the robosuite paper?
7. You make the following claims in Section 4.3: "[The LLE constraint] (1) encourages the learned embedding to preserve local neighborhood relationships reflecting the true physical dynamics [and (2)] disentanglement of dynamics-specific features from reward-specific features, improving interpretability and modularity." However, it seems no evidence was presented for these claims. E.g., Figures 7, 8 and 9 and the accompanying discussion do not seem to be direct evidence of any structure learning or neighborhood-preserving features. Given how central these claims are in your framing, this lack of direct evidence is surprising.

**Requested Changes:**

- Clarify the questions above and add clarifying or missing information into the manuscript, as appropriate.

- Further strengthen Section 3 by adding additional details:
	1. (In equation 1) What is the domain of the indexed variable $x_i$?
	2. (In 3.1.1) Similarly, where are the k-nearest neighbors chosen from?
	3. (In equation 2) What is the relationship of Equation 2 and $z_i^{\text{LLE}}$?

- Made the main text self-contained. In particular, it is currently necessary to refer to the Appendix because without Algorithms 1 and 2, the description of the framework is missing key details (e.g., the order in which the different steps are taken and the update of the network parameters on $L_RL$). Either move the algorithms to the main text or substantially strengthen the natural language description of the framework.

- Part of the framing and claims include "this framework generalize and make informed decisions based on both environment structure and reward signals." Thus, I suggest you discuss relevant structure-learning work currently missing:
	- Graphical models like rSLDS (Linderman et al, 2020). These works explicitly model environment structure by decomposing systems into a collection of simpler units.
	- DeepSynth (Hasanbeig et al, 2021). This work also incorporates the learning of dynamic-specific features, explicitly encoding environment structure into a discrete automata that drives the RL process.
	- SWMPO (Hernandez Cano et al, 2025). This work explicitly encodes environment dynamics structure into a neurosymbolic world model that can also be used within a RL process.

- Expand the description of the SAC-Next baseline. This baseline seems particularly relevant (arguably, it appears that this baseline also learns a representation of the state that includes information of the dynamics), but the description is far too short to appreciate the significance of the results.

- Please briefly describe in the main text the method for choosing the hyperparameters of your framework and the baselines.

---

> ### Author Response · Authors · 2026-01-16
>
> Thank you for the detailed and rigorous review. We appreciate the depth of your questions, particularly around the precise meaning of the LLE objectives, what we mean by “dynamics-specific,” the neighbor construction, and the relationship between our experimental setup and prior Robosuite results. Your comments also highlighted where our evidence and framing needed to be more direct (especially regarding neighborhood preservation and disentanglement). Below we respond point-by-point and indicate where the paper was updated.
>
> ---
>
> ## 1) Relationship between Eq. (2) and $z^{\mathrm{LLE}}$
>
> Eq. (2) enforces that **each embedded point** $z_i^{\mathrm{LLE}}$ is well-approximated by the **same neighbor-weighted reconstruction** used in the original space (via the weights from Eq. (1)). Concretely:
>
> - Original-space reconstruction weights:
>   - Given sampled states $\{x_i\}$ (where $x_i$ denotes the observation/state vector), for each $i$ we compute scalar weights $w_{ij}$ over neighbors $j \in \mathcal{N}(i)$ such that
>     - $x_i \approx \sum_{j \in \mathcal{N}(i)} w_{ij} x_j$
>
> - Embedding-space neighborhood preservation:
>   - The embedding $z^{\mathrm{LLE}}(x)$ is trained so that the same $w_{ij}$ reconstruct locally in the embedding:
>     - $z_i^{\mathrm{LLE}} \approx \sum_{j \in \mathcal{N}(i)} w_{ij} z_j^{\mathrm{LLE}}$.
>
> This is exactly the neighborhood-preservation mechanism of LLE: **weights computed in observation space are reused to preserve local structure in embedding space**.
>
> **Paper Update:** We have added two sentences explaining this after Equation (2).
>
> ---
>
> ## 2) “Dynamics-specific features” vs “state-wise information only”
>
> The embedding network is indeed **state-to-embedding** (it maps a single state/observation $x_t$ to $z_t^{\mathrm{LLE}}$), and we do not claim to explicitly learn a transition model $p(x_{t+1} \mid x_t, a_t)$.
>
> What we mean by “dynamics-specific” is:
>
> - The LLE constraint is applied to **states sampled from trajectories generated by the policy**, and neighbors are chosen as **trajectory-local (temporal) neighbors**.
> - Because the replay buffer contains states visited under the environment’s transition dynamics, enforcing local neighborhood consistency over **trajectory-local neighborhoods** encourages the embedding to be smooth and locally consistent along feasible transitions.
> - This yields a representation shaped primarily by **local geometry induced by the environment’s evolution** (what states are reachable and how they change locally), rather than being shaped directly by reward gradients.
>
> So, “dynamics-specific” here means: **the representation is learned using constraints derived from local structure of visited trajectories (policy-induced state visitation), rather than reward/return signals**. It is not a claim that we explicitly model dynamics with a predictive model, and we state this distinction clearly in Section 3.1.1 in the final paragraph.
>
> ---
>
> ## 3) Why physical dynamics make the “locally linear” assumption reasonable
>
> We do **not** assume global linearity or reversibility. The locally linear assumption used by LLE is:
>
> - In a **small neighborhood** around a state, points can be approximated as linear combinations of their neighbors.
> - This is an inductive bias about **local geometry**, not a claim that the dynamics are linear or that mappings are invertible.
>
> Why this is plausible in these domains:
>
> - Many robotics simulators evolve with **small integration steps**, and for small variations in state (and often action), the state space is locally smooth except around discontinuities (e.g., impacts/contact mode changes).
> - Our neighbor construction is **trajectory-local**, meaning neighbors tend to be close in time and therefore close in state. This increases the chance that local linear reconstruction is a reasonable approximation.
>
> We also acknowledge limitations: contact discontinuities can violate smoothness locally; in those cases the constraint is an inductive bias that may be imperfect.
>
> ---
>
> ## 4) Eq. (1): domain of the indexed variable $i$
>
> $i$ indexes elements of the **minibatch sampled from the replay buffer** at an LLE update step.
>
> - Let a sampled minibatch be $\{x_i\}_{i=1}^{B}$ where $B$ is minibatch size and $x_i \in \mathbb{R}^{D}$ is a state/observation vector.
> - Then $i \in \{1,2,\dots,B\}$.
> - Neighbors are selected as $\mathcal{N}(i)$ for each $i$ (see next point), and weights $w_{ij}$ are computed only for $j \in \mathcal{N}(i)$.
>
> So Eq. (1) is computed **over the minibatch**, not over all replay buffer entries.
>
> ---

---

> > ### Author Response · Authors · 2026-01-16
> >
> > ## 5) Where are the $k$-nearest neighbors chosen from?
> >
> > Neighbors are selected **in observation/state space** using **trajectory-local temporal neighbors** rather than a global kNN over the entire replay buffer. Concretely:
> >
> > - For a sampled transition at time index $t$ from a trajectory, neighbors are selected from a small temporal window around it, e.g., $x_{t-K/2}, \dots, x_{t-1}, x_{t+1}, \dots, x_{t+K/2}$.
> > - This gives a neighbor set $\mathcal{N}(i)$ consisting of states that are close in time (and typically close in state), which is efficient and consistent with the idea of preserving local geometry along experienced trajectories.
> > - Nearest-neighbor distances are computed directly in the observation/state space (not in latent space).
> >
> > This design avoids the computational burden of global neighbor search over a massive replay buffer while still enforcing local structure preservation.
> >
> > **Paper Update:** We have added this clarification in Section 3.1.1
> >
> > ---
> >
> > ## 6) Robosuite baselines appear much lower than in the Robosuite paper
> >
> >
> > The discrepancy came from a **training mismatch** between (a) our initial SAC implementation/training schedule and (b) the specific SAC implementation/training regime used in the Robosuite reference code used to generate their published curves. In particular:
> >
> > - Our initial experiments used a CleanRL-style SAC training schedule (frequent incremental updates), while the Robosuite reference implementation uses a different data collection and update regime (e.g., collecting a larger chunk of interaction and performing a block of many gradient updates), and also differs in the environement version and horizon handling (500 in robosuite 1.0 on which the repo was built vs 1000 which is the default in the latest version of robosuite).
> > - These implementation/training regime differences can substantially change learning curves even when both are labeled “SAC.”
> >
> > What we did to resolve this:
> >
> > - For **Robosuite**, we reran experiments using the SAC variant/protocol aligned with the Robosuite repository and updated the Robosuite results accordingly (and kept earlier plots only for transparency/traceability).
> > - For **Dexterous Gym**, the Robosuite SAC variant did not transfer without additional domain-specific tuning (it produced ~0 reward across tasks), so we retained the DexGym SAC setup that is stable in that environment. This avoids conflating “implementation mismatch” with “method effectiveness.”
> >
> > Interpretation guidance:
> >
> > - The key comparison is between our method and baselines **under the same protocol** per domain. When we align with the Robosuite protocol on Robosuite, baseline performance increases relative to the earlier draft, and the relative gains of the proposed representation can be evaluated fairly.
> >
> > **Paper Update:** We added an explicit paragraph in the experimental section stating which SAC protocol is used per domain and why.
> >
> > ---
> >
> > ## 7) Missing direct evidence for neighborhood preservation and disentanglement claims (Sec. 4.3)
> >
> > We agree that attention maps alone are not sufficient evidence of neighborhood preservation. We therefore provide direct quantitative evidence based on the reconstruction objective:
> >
> > - For each point we find its $K$ nearest neighbors in the original observation space, solve a least‑squares reconstruction of that point’s embedding from the neighbors’ embeddings, and report the reconstruction MSE in embedding space on 10 held-out evaluation rollouts across 10 seeds and compare SAC vs. SAC-LLE. We observed that **SAC-LLE yields substantially lower reconstruction error** than the SAC baselines. This indicates that the LLE-regularized representation preserves locally consistent structure more reliably in practice, providing direct support for our claim that the LLE constraint encourages neighborhood-preserving embeddings.
> >
> > We also clarify the disentanglement claim:
> >
> > - “Disentanglement” here is meant in the limited, operational sense that:
> >   - the dynamics branch is trained primarily by neighborhood-structure constraints (reward-independent),
> >   - the reward branch is trained by RL objectives (reward-driven),
> >   - and fusion selects between them per state.
> > - It is not meant as a claim of formal independent factorization without additional causal/identifiability assumptions.
> >
> > **Paper Update:** We added Section 4.5 emperically showing this claim.
> >
> > ---

---

> > > ### Author Response · Authors · 2026-01-16
> > >
> > > ## 8) Make main text self-contained (Algorithms 1 and 2 dependency)
> > >
> > > Here is the complete update order and parameter update rule:
> > >
> > > 1. Interact with the environment using the current policy and add transitions $(x_t, a_t, r_t, x_{t+1})$ to the replay buffer.
> > > 2. Perform standard SAC updates using minibatches from the replay buffer:
> > >    - Update critic(s) and actor with standard SAC losses.
> > >    - Update the reward-specific representation pathway as part of the SAC computation graph.
> > > 3. Periodically (every fixed interval), perform an LLE update step:
> > >    - Sample a minibatch of states $\{x_i\}_{i=1}^{B}$ from the replay buffer.
> > >    - Construct neighbor sets $\mathcal{N}(i)$ using temporal neighbors (trajectory-local).
> > >    - Compute LLE reconstruction weights $w_{ij}$ in observation space for each $i$ over $j \in \mathcal{N}(i)$.
> > >    - Update only the dynamics-embedding network parameters to minimize the neighborhood-preservation objective and Eq. (3).
> > > 4. Compute fused representation via the fusion mechanism and feed it to the policy and critic networks.
> > >
> > > This describes the key missing details (order of steps, which parameters are updated by which losses) directly.
> > >
> > > **Paper Update:** We have completely rewritten Section 3.2 to give more context to the training method.
> > >
> > > ---
> > >
> > > ## 9) Missing relevant structure-learning related work (rSLDS, DeepSynth, SWMPO)
> > >
> > > Thank you for pointing out these closely related structure-learning works. We have added a short discussion of them in the Related Works section. In brief, rSLDS, DeepSynth, and SWMPO explicitly represent environment structure via discrete regimes/automata/world models. In contrast, our approach does not infer an explicit structured dynamics model; instead, it introduces a lightweight **representation-level inductive bias** that preserves local neighborhood geometry (via LLE) and fuses it with reward-driven features through adaptive gating. This makes our method complementary to explicit structure-modeling approaches and potentially combinable with them.
> > >
> > > ---
> > >
> > > ## 10) SAC-Next baseline
> > >
> > >
> > > SAC-Next augments SAC with an auxiliary next-step prediction objective: it trains a predictor that takes the current representation and predicts a target related to $x_{t+1}$, encouraging features that are predictive of **transition dynamics**. This notion of “dynamics” is related to, but distinct from, what we mean by “dynamics-specific” in our LLE branch. In our method, “dynamics-specific” refers to **local dynamics / local geometric structure** induced by trajectories i.e., preserving local neighborhood relationships in the state manifold via the LLE constraint rather than explicitly modeling $x_{t+1}$ from $(x_t, a_t)$.
> > >
> > > Empirically, when next-step prediction is performed directly in the *vectorized observation/state space* (as in our SAC-Next baseline), the auxiliary gradients provide a relatively weak learning signal, leading to limited information propagation through this loss. In contrast, when prediction is performed in a learned *latent space* (SPR-style objectives), the auxiliary task is better conditioned and yields substantially stronger improvements, which is consistent with the improved performance of SPR-based baselines in our results.
> > >
> > >
> > > ---
> > >
> > > ## 11) Hyperparameter selection method (framework + baselines)
> > >
> > > LLE hyperparameters (e.g., LLE update interval, neighbor window size) are selected based on a small sensitivity sweep (highglighted in Section B of Appendix) and then held fixed across tasks (to avoid per-task tuning advantage).
> > >
> > > For baseline methods, we follow established configurations to avoid introducing tuning advantages. On Robosuite, we use the same SAC hyperparameters and training protocol as in the Robosuite reference implementation/paper for all baselines and our method, ensuring a direct and fair comparison. On Dexterous Gym, we use the default SAC hyperparameters from the CleanRL SAC repository for all baselines and our method, since this configuration is stable and commonly used in that setting. In both domains, hyperparameters are held fixed across methods within the same environment suite.
> > >
> > >
> > > ---

---

### Review · Reviewer_vpSf · 2026-01-04

**Summary Of Contributions:**

This paper proposes a dual-representation reinforcement learning framework that explicitly separates dynamics-specific and reward-specific state representations. The dynamics-specific branch uses Locally Linear Embedding (LLE) to capture local geometric structure in the state space, while the reward-specific branch is learned through standard RL objectives. A self-attention mechanism adaptively fuses these two representations on a per-state basis. The results validate that framework not only improves learning efficiency but also enhances overall performance with interpretable attention maps.nani

**Audience:**

Yes

**Audience Explanation:**

Yes. The paper should be of interest to readers working on reinforcement learning, representation learning, and biologically inspired ML.

**Broader Impact Concerns:**

/

**Claims And Evidence:**

Yes

**Claims Explanation:**

The claims are generally supported by experimental results. The proposed method consistently outperforms multiple auxiliary-loss baselines significantly based on the learning (average return) curve standard deviation bars across diverse robotic tasks, and ablation studies highlight the importance of decoupled learning and adaptive fusion. However, the relationship with neuroscience is marginal.

**Requested Changes:**

* The most significant improvement should be on the method presentation. Symbols/notations should be clearly defined, like the variable $z$, $s$, and how they are computed by the function, even it is trivial. Especially when your $z$ is locally embedded by some constraints. We need to have a clear meaning definition of $z$ and then understand its functionality to the whole system.
* Hyperparameters and experimental setup should be clearly explained. For example, how to train the model? The learning curve with standard deviation is based on how many random seeds. Also regarding the key LLE part, how the LLE update during the training procedure specifically? Are you minimizing the sum of a lot of loss terms jointly? Do they have corresponding term weights?
* I would suggest tone down or better contextualize neuroscientific claims to distinguish inspiration from validation.
* As far as my knowledge, finding nearest neighbors are computationally heavy. The local manifold construction for one datapoint is usually quadratic, resulting in a very expensive computation for the whole dataset. It is better to discuss computational overhead and scalability more explicitly.

---

> ### Author Response · Authors · 2026-01-16
>
> Thank you for the thoughtful review and for the balanced assessment of the paper’s contributions and limitations. Your feedback led us to strengthen the notation/definitions, standardize experimental reporting (seeds, uncertainty bands, training returns), and add clearer discussion of the LLE update schedule and computational overhead. We address each requested change below.
>
> ---
>
> ## 1) Method presentation: clarify symbols, notation, and definitions
>
> We agree, and we revised the method section to be fully self-contained by explicitly defining all variables, functions, and tensor conventions used throughout the pipeline.
>
> - **Made “locally embedded” explicit** by stating that LLE uses neighbor sets $\mathcal{N}(i)$ and reconstruction coefficients $w_{ij}$ computed in observation space (Eq. 1), and then enforces neighborhood preservation in the embedding by reusing the same coefficients to reconstruct $z_i^{\mathrm{LLE}}$ from its neighbors (Eq. 2). This clarifies the precise relationship between Eq. 2 and $z^{\mathrm{LLE}}$.
>
> - **Clarified domains and shapes inline** at the point of use. For example, we explicitly state $x_i\in\mathbb{R}^{D}$, $z_i^{\mathrm{LLE}}\in\mathbb{R}^{d}$, so concatenation/fusion operations are better explained.
>
> - **Removed notation ambiguity** by disambiguating neighbor weights $w_{ij}$ from any reconstruction/decoder parameters used elsewhere (e.g., Eq. 3), and by standardizing symbols across subsections (states/observations, embeddings, fused features).
>
> **Paper update:** These edits appear at the start of Sec. 3, and as additional inline clarifications around Eqs. (1)–(2) (relationship to $z^{\mathrm{LLE}}$) and Sec. 3.1.3 (fusion notation/shapes).
>
> ---
>
> ## 2) Hyperparameters and experimental setup: seeds, training protocol, and reporting
>
> - **Training protocol:** We rewrote Sec. 3.2 to explicitly describe the interleaved training process.
> - **Number of seeds:** The number of random seeds is stated in the main text (Sec. 4, Experimental Setup), and we now also include it in the caption of every figure.
> - **Uncertainty bands:** We clarify that the shaded regions in all learning curves indicate the **standard error across seeds** (not standard deviation), and we include this in figure captions.
> - **What return is shown:** The reported curves correspond to **training returns** (i.e., returns collected during training), and we now state this consistently in all relevant figure captions.
> - **Hyperparameters:** All remaining details required to reproduce training (optimizer settings, replay buffer parameters, update schedule, policy update frequency, batch sizes, etc.) are specified in the hyperparameter table in the Appendix. We will also make the provided code in the supplementary public.
>
> ---
>
> ## 3) LLE update procedure: when and how it is optimized, and loss weighting
>
> The LLE component is trained **periodically** using minibatches sampled from the replay buffer, and it is **not** optimized as an unstructured “sum of many losses” at every environment step. Concretely:
>
> ### When do LLE updates occur?
> - During training, we alternate between standard SAC updates and LLE updates.
> - **LLE updates are performed at a fixed interval** (every $M$ environment steps / SAC update steps) using a minibatch sampled from the replay buffer.
> - This means the LLE objective is applied **periodically**, not continuously at every interaction step.
>
> ### How are neighbors selected for LLE?
> - Neighbors are computed **in the observation/state space**.
> - We use **temporal neighbors**: for each sampled transition at time $t$, the neighbor set is constructed from nearby time indices (e.g., states from the same trajectory within a small temporal window). This avoids expensive global kNN over the full replay buffer while still enforcing local structure preservation on visited trajectories.
>
> ### What gets optimized in each update?
> - **SAC update (standard RL objective):**
>   - Updates: critic parameters and policy parameters (and the reward-specific branch if it is part of the policy/critic feature pipeline).
>
> - **LLE update (dynamics-specific objective):**
>   - Updates: **dynamics-specific embedding network parameters only** (the branch producing $z^{\text{lle}}(s)$).
>
> This separation ensures that the dynamics embedding is shaped primarily by neighborhood geometry constraints rather than reward gradients.
>
> ### Are the objectives minimized jointly or interleaved?
> - The training is **interleaved**, not fully joint at every step.
> - Therefore, we do **not** minimize “a single large weighted sum of SAC + LLE losses” at every iteration; instead, we alternate between two update modes with distinct objectives and parameter subsets.
>
> ---

---

> > ### Author Response · Authors · 2026-01-16
> >
> > ## 4) Neuroscience framing
> >
> > We have updated the paper to clearly separate neuroscience as **motivation** from what is **empirically validated** in this work. Concretely:
> > - All neuroscience-related statements are explicitly framed as **inspiration** (i.e., design motivation) rather than evidence-based claims about biological mechanisms.
> > - We state directly that our validation is based on **RL performance metrics** (returns, sample efficiency) and **representation-focused metrics/analyses** (e.g., neighborhood preservation and reconstruction behavior), not on neuroscientific experiments.
> >
> > ---
> >
> > ## 5) Computational overhead and scalability
> > We added an explicit compute and scalability discussion clarifying how neighbor selection and LLE computations are implemented efficiently in practice:
> >
> > - **No global neighbor graph is stored.** The LLE neighbor reconstruction weights $w_{ij}$ are computed **within minibatches** sampled from the replay buffer (or computed periodically), so storage is not $O(NK)$ over the full dataset. Instead, the storage cost is:
> >   - $O(BK)$ per LLE update, where $B$ is the minibatch size and $K$ is the number of neighbors.
> >
> > - **Neighbor selection is efficient and trajectory-local.** Rather than performing full kNN over the entire replay buffer, neighbors are selected as **nearby temporal states** (trajectory-local neighbors) in observation/state space. This avoids the expensive quadratic scaling that would arise from global nearest-neighbor search across all stored transitions.
> >
> > - **LLE updates are periodic.** The LLE objective is computed at a fixed update interval, so the neighbor computation overhead does not occur at every environment step.
> >
> > - **Overhead reporting:** We report training throughput overhead relative to SAC (e.g., average steps/sec) in Appendix Table 2 and describe memory overhead in practical terms in Section 3.1.1 (additional minibatch-level neighbor weights and any auxiliary tensors used during LLE updates).
> >
> > ---

---

### Review · Reviewer_8wCP · 2026-01-13

**Summary Of Contributions:**

This paper proposes a framework for Reinforcement Learning (RL) state representations that capture the underlying dynamics of the environment, without being tied to the reward function. Using the Locally Linear Embedding (LLE) framework to preserve local neighbourhood structures, dynamics-specific features are extracted, while reward-specific features are learned though conventional RL loss. The authors claim that experiment carried in a robotics simulation settings reveal that the method improves learning efficiency and performance compared to conventional RL.

The key strengths and weaknesses are listed below.
### Strengths
- The idea to leverage embeddings that capture the physics of the environment is an interesting one, heavily important in other areas of scientific machine learning.
- I find *unsupervised* LLE addition well-inspired, as the local linear structure in the latent space must mirror that of the original space.
- The authors make significant efforts to draw parallels with the functioning of the brain, which helps motivate the design of specific components.

### Weaknesses
- I find the mathematical presentation not convincing, as some vector and matrix shapes need clarification
- The evaluations are weak, and critical ablation studies missing. A toy dataset with known locally linear geometric structure would have immensely helped this work.

**Audience:**

Yes

**Audience Explanation:**

Among other strengths of this paper, the TMLR audience will be interested in how to incorporate other kinds of physical biases, in an unsupervised way, into the environment to improve efficiency and performance.

**Broader Impact Concerns:**

No concerns found.

**Claims And Evidence:**

No

**Claims Explanation:**

I am not an expert in neuroscience, and cannot speak as to the accuracy and convincingness of all claims relating to the brain analogies made by the authors. My assessment is purely based on the machine learning content of this paper.

Although the results show exceptional performance compared to several baselines in the soft actor critic robotic simulation setting. I am not convinced by these, since the paper lacks ablation studies to empirically support the choice of components. The evaluation scenarios are weak, and my points below better clarify my concerns.

**Requested Changes:**

### Critical
1. $f_{cat}$ in section 3.1.3 needs clarification. I assume it belongs to $\mathbb{R}^{2d}$? The current writing would suggest a horizontal concatenation of two column vectors, resulting in a matrix of shape $\mathbb{R}^{d\times 2}$. It should be corrected to involve transposition for clarity.
2. Following from the previous question, this would mean the output of the self-attention $f$ is of shape $\mathbb{R}^{2d}$ just like its input, and such a vector would be fed into a Action Selection/Policy network as per Figure 1. Presumably, $f_{cat}$ contains just as much information as $f$ (if not more). Could $f_{cat}$ be passed to the policy network directly? Please clarify the role of the Self-Attention through carefully crafted ablation studies.
3. In Eq (3), please specify the shapes of every element. Is $W_i$ the same reconstruction weights as learned in Eqs (1) and (2)? Also, on what basis is it assumed that the relation between the learned latent features and the original features is locally linear (I would assume this relation is generally non-linear, and no necessarily reversible? Finally, an ablation of the loss Eq (3) would reveal the core of the framework and help better understand the role of each component in Figure 1.


### Optional (would strengthen the work in my view)
4. The proposed SAC-LLE framework approach introduces at least $N_{datapoints}*N_{neighbors}$ additional weights to learn, whose repercussions beyond expressivity are not examined. What is the computational cost of adding an LLE module compared to a single conventional RL module that performs both tasks?
5. Comparison has only been performed in the Soft-Actor-Critic regime. It remains to be seen how this LLE approach performs in other scenarios.

---

> ### Author Response · Authors · 2026-01-16
>
> Thank you for the careful and detailed review. We appreciate the time you took to identify both the strengths of the submission and the specific points where the presentation and empirical evidence needed to be clearer. Your questions on notation (especially Sec. 3.1.3 and Eq. (3)), the role of the fusion mechanism, the missing ablations, and computational considerations directly helped us improve the paper. Below we respond point-by-point and summarize the concrete updates made.
>
> ---
> ## Critical 1 — Section 3.1.3: Clarify concatenation and tensor shapes
>
> We agree the current presentation is ambiguous. Our intent is **feature-based concatenation**, i.e., concatenation along the feature/channel dimension (not forming a matrix by horizontal concatenation of column vectors).
>
> In the revised manuscript, we explicitly define shapes. For example:
> - Dynamics-specific embedding: $z^{\mathrm{lle}}(s) \in \mathbb{R}^{d}$
> - Reward-specific embedding: $z^{\mathrm{rew}}(s) \in \mathbb{R}^{d}$
>
> Feature concatenation:
> $$
> f_{\mathrm{cat}}(s) = \mathrm{concat}\left(z^{\mathrm{lle}}(s),\, z^{\mathrm{rew}}(s)\right) \in \mathbb{R}^{2d}.
> $$
>
> **Paper change:** We added a clarification paragraph in Sec. 3.1.3 explicitly stating the above conventions to remove any ambiguity.
>
> ---
>
> ## Critical 2 — Why self-attention? Why not feed $f_{\mathrm{cat}}$ directly to the policy?
>
> We agree that $f_{\mathrm{cat}}$ contains at least as much information as the attended representation. However, there are two practical motivations for using a **gating/fusion mechanism (self-attention in our implementation)** rather than directly feeding the concatenation:
>
> **Adaptive Weighting:** Self-attention is not an extra flourish but the only learned fusion between the reward-stream features and the LLE features: we first concatenate the two 128-d streams and then reweight them with self attention. Dropping attention simply reduces to a static concat and removes the model’s ability to adaptively gate which stream matters per state.
>
> **Interpretability:** The attention mechanism provides **state-dependent mixing weights**, enabling inspection of which branch/features dominate per state (as visualized via attention maps). Pure concatenation does not naturally provide this mechanism for analyzing per-state feature dominance.
>
> We also agree that these motivations should be supported empirically.
>
> **Paper update:** We added Section 4.6 empirically justifying our chosen approach.
>
> ---
>
> ## Critical 3 — Eq. (3): Shapes, weight overloading, and meaning of “local linearity”
>
> We agree the notation is currently confusing. In particular, **$W$ is overloaded**: the reconstruction weights in the LLE objective and the parameters used in Eq. (3) are not the same.
>
> - In the LLE formulation (Eqs. (1) and (2)), the weights $w_{ij}$ are **scalar neighbor reconstruction coefficients** (typically constrained, e.g., sum-to-one) used to represent each point as a linear combination of its neighbors. These are not learnable “global model weights” and are not intended to be reused as a decoder.
>
> - In Eq. (3), we use a **separate reconstruction regularizer** to prevent degenerate embeddings and to encourage locally consistent structure. The parameters used here are **different** from $w_{ij}$. We renamed these parameters to avoid confusion and specify their shapes explicitly.
>
> Regarding “local linearity”: we do **not** assume global linearity or invertibility. The key LLE assumption is:
>
> > **Points in the embedding space can be approximated as linear combinations of their neighbors (locally).**
>
> This is a **local** property used as an inductive bias and does not imply:
> - a globally linear mapping,
> - global reversibility, or
> - exact reconstruction everywhere.
>
> **Paper change:** We rewrote Eq. (3) with unambiguous symbols and added a shape clarification for all components in Eq. (3), explicitly distinguishing neighbor weights $w_{ij}$ from reconstruction/decoder parameters.
>
> ---
>
> ## Critical 4 — Ablation of Eq. (3)
>
> Eq. (3) is the **reconstruction regularizer** applied to the LLE branch. While the neighborhood preservation objective (Eqs. 1–2) enforces that embeddings respect local linear neighborhood structure, Eq. (3) ensures that the learned embedding remains **information-preserving** and avoids degenerate solutions (e.g., collapsed or overly lossy embeddings) by encouraging the embedding to retain state-relevant information.
>
> We compare two variants that differ only in whether the reconstruction regularizer is applied:
>
> - **SAC-LLE:** LLE neighborhood preservation + Eq. (3) reconstruction regularizer.
> - **SAC-LLE (no regularizer):** LLE neighborhood preservation **without** Eq. (3).
>
> All other components (SAC updates, two-stream design, fusion mechanism, neighbor construction, and training schedule) remain unchanged.
>
> **Paper update:** We added Section 4.7 showing the necessity of the reconstruction regularizer; removing it results in a drop in performance.

---

> > ### Author Response · Authors · 2026-01-16
> >
> > ## Optional 1 — Compute cost and “additional weights” concern
> >
> > The LLE neighbor weights $w_{ij}$ are **not stored globally for all datapoints** as persistent parameters. Instead, they are computed **within minibatches** sampled from the replay buffer (or computed periodically), so the storage is:
> > $$
> > \mathcal{O}(BK)
> > $$
> > per update, where $B$ is the minibatch size and $K$ is the number of neighbors.
> >
> > Additionally, in our implementation the nearest neighbors are calculated **directly in observation space** based on the policy’s collected experience. In practice, we use **temporal neighboring states** as neighbors, which significantly reduces neighbor search overhead compared to a full kNN search across the buffer.
> >
> > **Paper update:** We added a compute/scalability paragraph reporting how neighbors are selected (temporal neighbors) and memory footprint implications in Section 3.1.1 of the revised paper. Additionally, we report compute comparisons across various baselines in Appendix Table 2.
> >
> > ---
> >
> > ## Optional 2 — Only evaluated under SAC
> >
> > We agree this is a limitation of the current experimental scope. We chose SAC because it is a strong and commonly used baseline in robotics continuous-control tasks. Importantly, nothing in the representation-learning formulation is SAC-specific: the learned representation and neighborhood constraint can be integrated into other RL algorithms as well.
> >
> > ---
> >
> > ## Optional 3 - Toy Environment
> >
> > We understand the reviewer’s suggestion that a toy dataset with analytically known manifold geometry can provide an additional sanity check. In our setting, however, Robosuite and Dexterous Gym already provide *structured, physically grounded state spaces* where local geometry is well-defined by robot kinematics and physics constraints. The observations are composed of structured quantities such as joint configurations/velocities and object poses, and transitions are generated by a physics simulator with small integration steps. As a result, the state visitation distribution along trajectories exhibits strong *local smoothness* (within a temporal neighborhood) except at contact-induced discontinuities. This is precisely the regime targeted by our LLE formulation, which operates on trajectory-local temporal neighbors and enforces locally linear reconstruction only within these neighborhoods (rather than assuming a globally linear or globally smooth manifold).
> >
> > While these benchmarks do not provide a single closed-form ground-truth manifold for the entire state distribution, they do offer a realistic and widely used testbed where the relevant assumption holds in practice. To further support this directly, we add quantitative neighborhood-preservation evidence via the reconstruction diagnostics (Eq. 3) across rollouts, showing that SAC-LLE yields substantially lower reconstruction error than SAC baselines, consistent with improved local-structure preservation.

---

### Decision · Action_Editor_U9ah · 2026-04-01

**Recommendation:** Accept as is

**Audience:**

Yes

**Audience Explanation:**

readers across neuroscience and reinforcement learning would be interested in this

**Claims And Evidence:**

Yes

**Claims Explanation:**

claims are supported by evidence. authors generally lean accept